# Structural basis of RNA polymerase inhibition by viral and host factors

Simona Pilotto [1], Thomas Fouqueau[1], Natalya Lukoyanova [2], Carol Sheppard [3], Soizick Lucas-Staat[4], Luis Miguel Díaz-Santín[2], Dorota Matelska [1], David Prangishvili[5], Alan C. M. Cheung [6] & Finn Werner [1]✉

RNA polymerase inhibition plays an important role in the regulation of transcription in response to environmental changes and in the virus-host relationship. Here we present the high-resolution structures of two such RNAP-inhibitor complexes that provide the structural bases underlying RNAP inhibition in archaea. The Acidianus two-tailed virus encodes the RIP factor that binds inside the DNA-binding channel of RNAP, inhibiting transcription by occlusion of binding sites for nucleic acid and the transcription initiation factor TFB. Infection with the Sulfolobus Turreted Icosahedral Virus induces the expression of the host factor TFS4, which binds in the RNAP funnel similarly to eukaryotic transcript cleavage factors. However, TFS4 allosterically induces a widening of the DNA-binding channel which disrupts trigger loop and bridge helix motifs. Importantly, the conformational changes induced by TFS4 are closely related to inactivated states of RNAP in other domains of life indicating a deep evolutionary conservation of allosteric RNAP inhibition.

[1] RNAP Laboratory, Institute for Structural and Molecular Biology, University College London, London, UK. [2] Institute for Structural and Molecular Biology, Birkbeck College, London, UK. [3] Section of Virology, Department of Infectious disease, Imperial College London, London, UK. [4] Department of Microbiology, Institut Pasteur, Paris, France. [5] Ivane Javakhishvili Tbilisi State University, Tbilisi, Georgia. [6] School of Biochemistry, University of Bristol, Bristol, UK. ✉email: f.werner@ucl.ac.uk

RNA polymerases (RNAPs) are the engines of transcription and an important target for the regulation of gene expression relevant for health and disease. Gene-specific factors regulate transcription dependent on promoter DNA sequence motifs, while global repression or attenuation of all RNAs is enabled by regulators that directly interact with RNAP independently of the template DNA[1,2]. The underlying mechanisms of RNAP inhibition are not only of academic interest but also in the context of the emerging antibiotic resistance crisis, since many frontline antibiotics including Rifampicin are RNAP inhibitors[3]. By exploring the structural bases of RNAP inhibition we not only learn about the mechanisms of transcription, but also inform the rational design of novel antibiotics[4]. In this context, viral transcription factors that inhibit the host RNAP are a rich hunting ground. Viruses are the most abundant pathogens found in nature and important drivers of evolution because of their ability to facilitate horizontal gene transfer[5]. Once viruses have infected a cell, they are subjected to similar needs: to bypass cellular immunity and to rewire the host gene expression machinery for their own benefit—chiefly to produce virus particles.

Well studied bacteriophages like T7 prevent the expression of host genes by inhibiting the host transcription machinery, and replace it with virus-encoded components that exclusively transcribe viral genes. The bacteriophage T7 factors Gp2 and P7 target the RNAP β′ subunit directly and not only interfere with σ[70] binding but also alter the conformational dynamics of RNAP during open complex formation and DNA melting, an essential step during early transcription initiation[6]. Inhibition of host transcription is an obvious strategy for viruses that encode their own RNAP, but archaeal viruses are entirely dependent on the host transcription machinery, as their genomes do not encode recognisable RNAP subunit genes. Hence, viral promoters utilise host promoter motifs which are active in in vitro transcription experiments using host RNAP and general initiation factors[7]. We have recently characterised the molecular mechanisms of the highly toxic RNAP inhibitor RIP (RNAP Inhibitory Protein, aka ORF145) encoded by the Acidianus two-tailed virus (ATV) that infects crenarchaea including *Sulfolobus* species[7]. RIP has undergone a fascinating functional diversification; it is evolutionary related to a capsid protein ORF131, but it binds RNAP with high affinity and efficiently inhibits transcription initiation and elongation[7]. But the global inhibition of RNAP is not restricted to virally-encoded factors, as a plethora of cellular factors bind directly to and inhibit RNAP in response to unfavourable environmental conditions including stress. The host encoded and constitutively expressed Gfh1 (Gre-factor homologue-1)[8] and DksA (DnaK suppressor A)[9] are inhibitors of bacterial RNAP that enable the fine-tuning of transcription, and belong to the group of factors that act through the nucleotide triphosphates (NTPs) entry funnel of RNAP, or secondary channel[10]. These also include positive regulators such as the eukaryotic RNAPII TFIIS[11], archaeal TFS[12,13] and bacterial GreA/B[14,15] that enhance elongation and resolve stalled and backtracked transcription elongation complexes (TEC) by transcript cleavage. We have previously described the molecular mechanisms of the archaeal TFS paralogue TFS4 as RNAP inhibitor[16]. TFS4 destabilises RNAP–DNA complexes and inhibits catalysis by decreasing the affinity for NTP substrates[16]. Intriguingly, TFS4 cannot be detected in the cell during normal growth conditions, but it is strongly upregulated in response to viral infection with STIV (Sulfolobus Turreted Icosahedral Virus)[17].

We have applied single particle cryo-electron microscopy (cryo-EM) to characterise the structural bases of RNAP inhibition in archaea. Here, we present the high-resolution structures of the inhibitors RIP and TFS4 in complex with the 13-subunit RNAP from *Sulfolobus acidocaldarius*. Our results reveal intricate interaction networks between inhibitors and RNAP that rationalise their modes of action. RIP binds to the RNAP clamp inside the DNA-binding channel and interferes with transcription by occluding DNA and transcription factors binding sites. Intriguingly, the C-terminal tail of RIP interacts with the RNAP similarly to the B-linker of the basal factor TFB (homologous to TFIIB in eukaryotes). In contrast, TFS4 interacts with RNAP through the funnel. Rather than interfering with the binding of transcription factors or nucleic acids, TFS4 inhibits RNAP by inducing large-scale conformational changes that result in the opening of the DNA-binding channel, and the disruption of active site motifs including the bridge helix and the trigger loop. As similar structural perturbations occur in other inhibited states of RNAP, our results demonstrate that the allosteric inhibition of the RNA polymerase is evolutionary conserved across all domains of life.

## Results

**The cryo-EM structure of the Saci apo-RNAP.** We first determined the cryo-EM structure of the apo-RNAP from *Sulfolobus acidocaldarius* (Saci) at 2.9 Å resolution (Table 1 and Supplementary Fig. 1).

The Saci RNAP consists of 13 subunits (Rpo1–13, Fig. 1a) with a molecular weight of 405 kDa. The active site environment is well resolved, with a clearly defined bridge helix, trigger loop, and the catalytic aspartate triad (see below). The overall structure displays a strong correspondence both at sequence and structure level to crenarchaeal *S. solfataricus*[18] and *S. shibatae*[19] RNAPs and eukaryotic RNAPII[20] (Supplementary Figs. 2 and 3).

The Saci RNAP structure includes eight conserved metal centres, six zinc coordinating motifs, a redox-inactive[18] cubane [3Fe-4S] iron-sulphur cluster, as well as the catalytic magnesium ion (Mg[A]) that is characteristic for multisubunit, double-psi beta barrel (DPBB) RNAPs (Fig. 1b and Supplementary Figs. 2, 3, and 4a). The high resolution enabled us to identify five prolines in the *cis* configuration and an additional zinc finger domain not previously reported (Supplementary Fig. 4a). Compared to previously published archaeal RNAPs, our cryo-EM map revealed a sequence register issue in the Rpo8 model starting from loop β5–6. Interestingly, the chain retracing led to the repositioning of the conserved GGLLM motif[21] at the interface with Rpo1′, as seen in all RPB8 subunits of eukaryotic RNAPI, II, and III (Fig. 1c and Supplementary Methods and Supplementary Fig. 4b, c).

**ATV RIP adopts a five-helical bundle structure.** In order to gain insights into the structural basis of RIP function and to identify the accurate binding mode of RIP, we solved the cryo-EM structure of the RNAP–RIP complex (Fig. 2b). A 10-fold excess of bacterially expressed recombinant RIP was incubated with Saci RNA polymerase and crosslinked with BS[3] (see "Methods" section). BS[3] is a mild crosslinker that showed lower propensity to induce aggregation and precipitation compared to other known crosslinkers during complex reconstitution carried out at 65 °C. Data were collected using the same approach used for the apo-RNAP and the processing provided a map with a final map resolution of 3.3 Å (Fig. 2 and Supplementary Fig. 5, statistics reported in Table 1).

The amino acid sequence of RIP suggests that it is a paralog of the viral P131/ORF131 capsid protein[7]. The ancestral gene of ORF131 underwent gene duplication and speciation and the RIP paralogue acquired the novel function of tightly binding to RNAP and inhibiting its activity[7]. The structure of RIP confirms this close relationship, as both RIP and ORF131 adopt a highly superimposable coil bundle structure with six α-helices and a

**Table 1 Cryo-EM data collection and refinement statistics.**

| | apo-RNAP | RNAP/RIP | RNAP/TFS4 | Stalk[a] |
|---|---|---|---|---|
| Data collection and processing | | | | |
| Microscope | Titan Krios | Titan Krios | Titan Krios | – |
| Voltage (kv) | 300 | 300 | 300 | – |
| Detector | K3 | K2 Summit | K3 | – |
| Electron exposure (e/Å²/frame) | 1.1115 | 1.01 | 1.13 | – |
| Defocus range (μm) | 1.3–2.5 | 1.5–3.5 | 1.5 - 3 | – |
| Data collection mode | Super-resolution 30° tilt | Counting 30° tilt | Super-resolution | – |
| Physical pixel size (Å/pixel) | 1.085 | 1.047 | 1.085 | – |
| Symmetry imposed | C1 | C1 | C1 | – |
| Initial particle images | 1,286,432 | 600,640 | 1,161,535 | – |
| Final particle images | 423,157 | 151,237 | 350,682 | – |
| Map resolution (Å)[b] | 2.88 | 3.27 | 2.61 | 3.75 |
| Map sharpening B-factor (Å²) | −57.7772 | −81.4405 | −32.6741 | −81.6388 |
| Refinement | | | | |
| Model composition | | | | |
| Chains | 13 | 14 | 14 | 2 |
| Residues | 3234 | 3373 | 3305 | 290 |
| Ligands | 1 Mg | 1 Mg | 1 Mg | none |
| | 6 Zn | 6 Zn | 8 Zn | |
| | 1 3Fe-4S | 1 3Fe-4S | 1 3Fe-4S | |
| Map to model cc score | 0.85 | 0.84 | 0.83 | 0.81 |
| Molprobity statistics | | | | |
| Clashscore | 5.66 | 9.58 | 4.67 | 9.82 |
| Ramachandran favoured (%) | 96.50 | 97.00 | 97.12 | 95.80 |
| Ramachandran outliers (%) | 0.00 | 0.00 | 0.00 | 0.00 |
| Bond lengths (%) | 0.03[c] | 0.03[c] | 0.03[c] | 0.00 |
| Bond angles (%) | 0.04[c] | 0.04[c] | 0.04[c] | 0.00 |
| Rotamer outliers (%) | 0.00 | 0.00 | 0.00 | 0.00 |
| Cβ outliers (%) | 0.00 | 0.00 | 0.00 | 0.00 |
| Molprobity score | 1.54 | 1.68 | 1.40 | 1.80 |
| CaBLAM outliers (%) | 2.2 | 1.4 | 1.9 | 2.5 |
| CA geometry outliers (%) | 0.69 | 0.39 | 0.59 | 1.42 |

[a]From multibody refinement of RNAP/TFS4 dataset.
[b]Gold-standard FSC 0.143 cutoff criteria.
[c]It refers to the Fe₃S₄ cluster.

rmsd of 1.7 Å (Fig. 2a). We prepared an improved sequence alignment of RIP/ORF131 paralogues, informed by our cryo-EM structure, which allowed us to identify the sequence determinants and the structural features responsible for the functional specialisation of RIP (Fig. 2a). The alpha-helical bundle of RIP is connected to a RIP-specific long C-terminal tail that is not conserved in any of the ORF131 paralogues (Fig. 2a)[7]. We could resolve RIP encompassing residues 10–127 (of 145 aa) within the RNAP–RIP complex, and the secondary structure content of RIP is in good agreement with CD-spectra recorded with free RIP which predicted a predominantly alpha-helical structure[7]. The first five α-helices are highly conserved between RIP and ORF131. The insertion of three additional residues, L99–D100–T101, allows the α6* helix to fold partially back on α5, and project the C-terminal tail inside the DNA-binding channel of RNAP. The difference in the position and orientation of the sixth α-helix is a crucial feature that allows the tail of RIP to adopt the correct position required for its interaction with RNAP (Fig. 2a, c).

**RIP forms a plug in the DNA binding channel of RNAP.** Our recent cross-linking/mass spectrometry studies suggested that RIP interacts with the RNAP clamp inside the DNA-binding channel of the archaeal RNAP[7]. The cryo-EM structure of the RNAP–RIP complex provides the detailed structural basis for this interaction. The compact structure of RIP allows it to fit snugly in the DNA-binding channel of RNAP between the Rpo1′ clamp, and the Rpo2 protrusion and lobe motifs, respectively, on each side of the

channel (Fig. 2b). The C-terminal tail of RIP forms an intricate network of interactions with the RNAP clamp and rudder motifs with an interface area of 1127 Å² that includes both hydrogen bonds and hydrophobic interactions (Fig. 2c and Supplementary Fig. 7a, c). The RIP tail makes a 90° bend at residue N117 and the region L114–M119 adopts an unusual L-shaped conformation that fits into a pocket between rudder and clamp (Fig. 2c). On the opposite side of the DNA binding channel, RIP interacts with the RNAP Rpo2 protrusion and lobe motifs mainly via hydrophobic interactions and with a small interface area of 302 Å². Taken together, the tight interaction network between RIP and RNAP provides a structural rationale for the extreme salt-resistant binding that persists at up to 2 M NaCl[7].

The extremity of the RIP tail (128–145) that was not resolved in the structure is enriched in negatively charged residues (6/18) that are not conserved in ORF131; these possibly mimic the negative charge of the DNA template phosphodiester backbone. The importance of the C-terminal tail for RNAP binding is in good agreement with the previously published EMSA data, which showed that a C-terminal truncation (RIP Δ114–145) abrogated the RNAP binding and inhibitory activities of RIP, while not compromising the extreme heat-stability of the protein[7]. Some of the critical residues of Rpo1′ that are involved in RIP binding including K238, R244, D241, H272, and R290 are strictly conserved among archaeal RNAPs but not with bacterial RNAPs (Supplementary Fig. 7d). This is in good agreement with the observations that (i) RIP also inhibits the euryarchaeal *M. jannaschii* RNAP in vitro, and (ii) the fact that

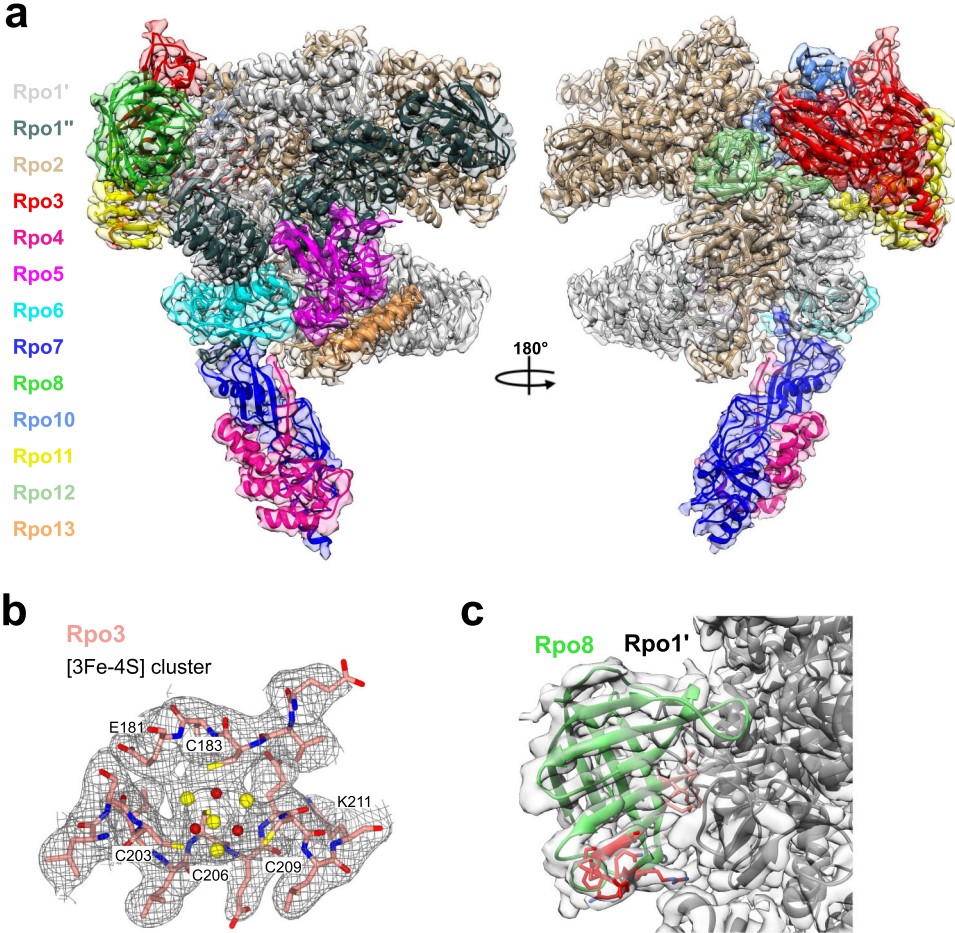

**Fig. 1 The complete structure of the crenarchaeal RNA polymerase. a** Cryo-EM map and cartoon representation of *Sulfolobus acidocaldarius* RNA polymerase. This composite map is illustrated as semi-trasparent surface and obtained by merging the cryo-EM map of apo-RNA polymerase with the stalk resulting from the multi-body refinement of the TFS4-bound RNA polymerase. Each subunit is highlighted according to the colour code shown on the left, the cryo-EM map is illustrated as semi-transparent surface. **b** Details of the apo-RNAP map showing the coordination system of the [3Fe-4S] cluster with sulphur atoms in yellow spheres and iron in dark red. Rpo3 residues are shown as sticks in red as in **a**, while the map is shown in mesh. **c** Cartoon representation of Rpo8 bound to Rpo1´ within the cryo-em map shown here in semi-transparency. The β5–6 loop is highlighted in red, and the conserved GGLLM motif in orange. The corresponding sequence alignment is reported in Supplementary Fig. 4b.

recombinant RIP can be overexpressed in large quantities in *E. coli* while being extremely toxic to *Sulfolobus acidocaldarius*[7]. The flexible RNAP clamp can adopt open and closed states which result in changes in the width of the DNA-binding channel in response to the engagement of RNAP with the DNA template, and binding of initiation and elongation factors, respectively[22]. Similarly to other DPBB RNAPs, the archaeal enzyme cycles between multiple distinct conformational states as RNAP progresses through the transcription cycle. We have previously applied smFRET to monitor the conformational changes of the *M. jannaschii* RNAP in solution, and found that RIP strongly favoured one fixed closed conformation of the clamp[7]. The RIP–RNAP interaction network that involves both sides of the RNAP DNA-binding channel provides a persuasive structural rationale for the nanomolar binding affinity and RIP's ability to lock the clamp in a fixed conformation[7].

**Structural determinants of RIP inhibition.** The structure and function of the archaeal preinitiation complex (PIC) is conserved with the eukaryotic RNAPII system; the combination of TBP and TFB is necessary and sufficient to enable RNAP recruitment and start site-specific transcription initiation at basal levels[23]. TFE activates transcription by inducing conformational changes in RNAP and enhancing DNA strand separation during the closed to open PIC transition[24,25]. Electrophoretic mobility shift assay (EMSA) experiments using the *Saccharolobus solfataricus* RNAP, and initiation factors demonstrated that RIP interferes with PICs by counteracting their formation and by destabilising preformed PICs[7]. A superposition of the RIP-RNAP structure with the closed PIC (from yeast, pdb 6gyk[20]) reveals that the C-terminal tail of RIP (residues 117–123) and the helix α4 (Fig. 2a) overlap with the TF(II) B residues 90–120 encompassing the B-linker and B-linker helix, respectively, by adopting the same structure and equivalent binding mode to the RNAP rudder (Fig. 3a, b). Consequently, RIP inhibits the PIC by competing with TFB binding to RNAP. In addition, RIP occludes the binding site to the nucleic acid scaffold in both closed and open PIC (pdb 5iyd[26]), and thus would prevent the loading of the DNA in the RNAP active site (Fig. 3c).

In contrast to the strong inhibition of initiation factor-dependent and promoter-directed transcription initiation, RIP has a smaller effect on elongation, and EMSAs showed that RIP, albeit in a limited fashion, can bind to TECs consisting of RNAP and a DNA/RNA scaffold[7]. Indeed, a superposition of the RNAP–RIP structure with a eukaryotic TEC (pdb 5oik[27]) reveals that the downstream duplex DNA, and the unwound template DNA strand passes underneath RIP while the non-template strand (NTS) clashes with RIP (Fig. 3f, g). To evaluate the

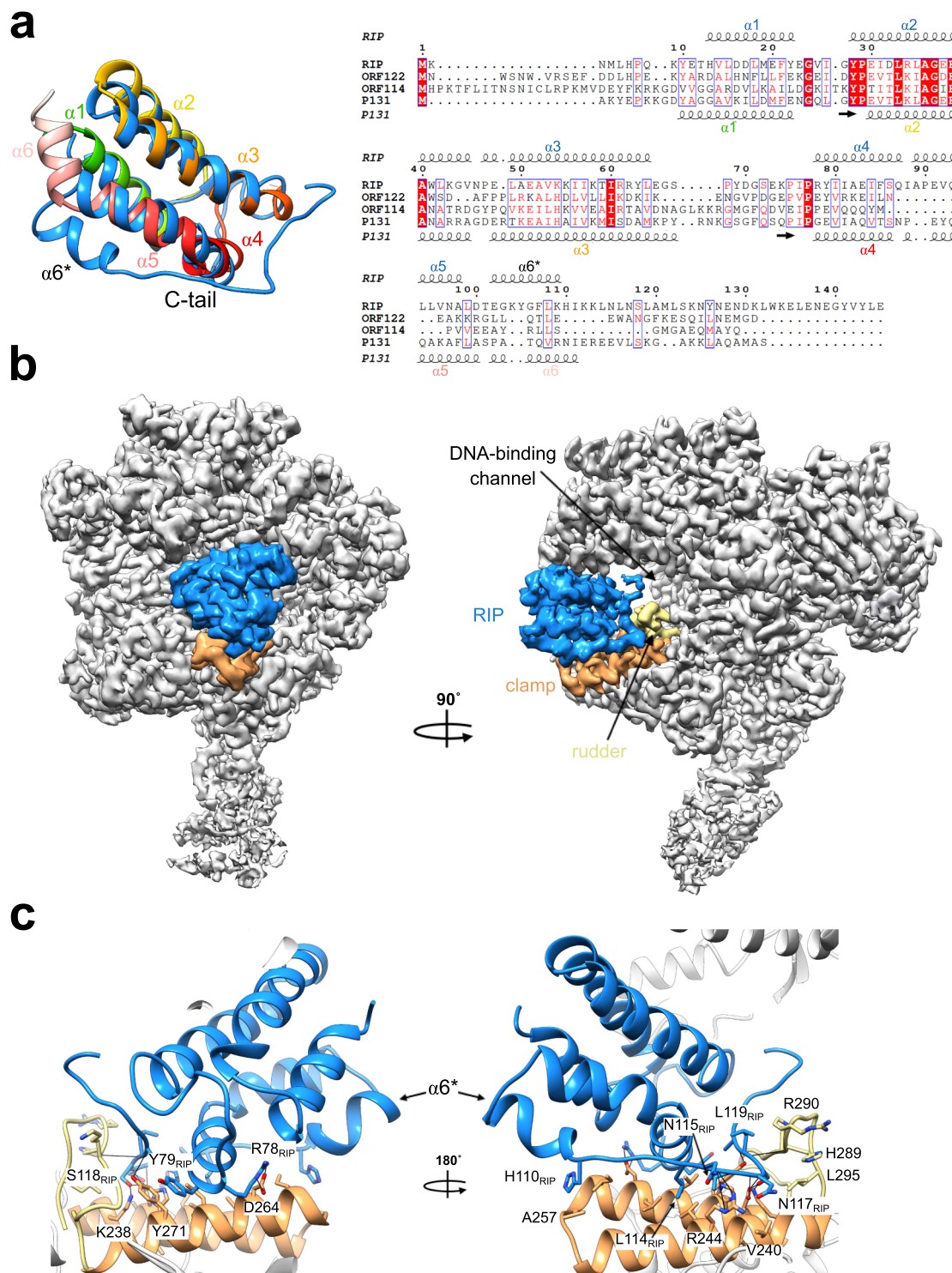

**Fig. 2 RIP forms a plug on top of the DNA binding channel of RNAP. a** Structure superimposition of RIP and its viral relative ORF131/P131 (pdb code 3faj). RIP is shown in blue and ORF131/P131 in a gradient of colours from the N-terminus in green to C-terminus in light pink with the helix numbering indicated. The orientation of the RIP α6 helix (indicated with an asterisk) diverges from P131. The corresponding multiple sequence alignment with viral proteins closely related to RIP was performed in PROMALS3D. They include SMV-1 ORF122 and ORF114, as well as the ATV P131 (ORF131). The secondary structure annotation generated by Espript3 for RIP and P131 are shown, respectively, above and below the sequence alignment. **b** The cryo-EM map of the Saci RNAP-ATV RIP complex reveals that RIP forms a plug in the DNA binding channel. RIP is coloured in blue, the RNAP clamp and rudder motifs in orange and yellow, respectively. **c** Enlarged view highlighting selected residues of RIP, RNAP clamp head and rudder motifs that are involved in the interaction shown in stick representation. The detailed RIP-RNAP interfaces analysis is illustrated in Supplementary Fig. 7a.

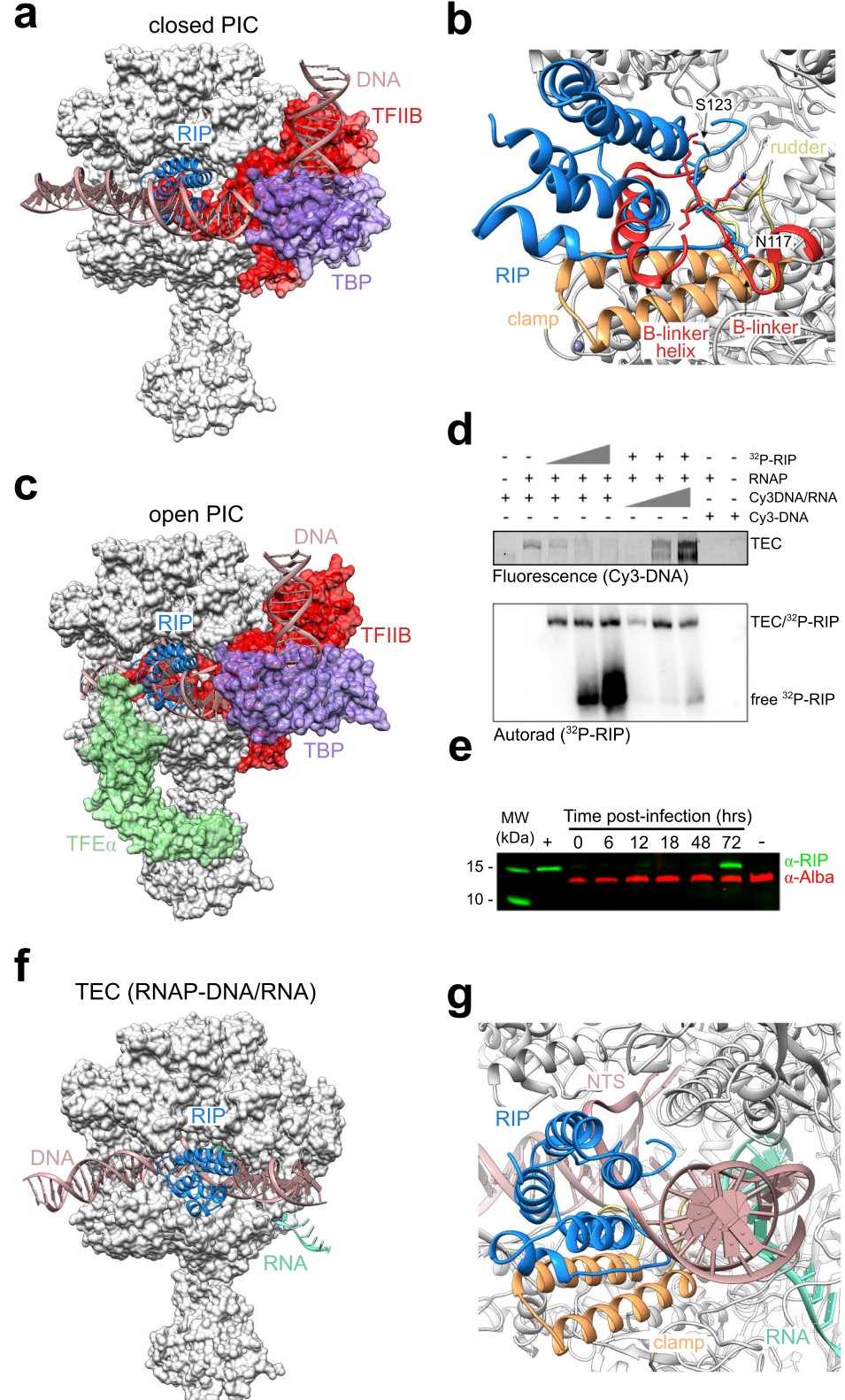

**a** closed PIC

**b**

**c** open PIC

**d**

**e**

**f** TEC (RNAP-DNA/RNA)

**g**

contribution of the NTS clash to the inhibition, we carried out EMSA experiments with dual labelled components; RIP was radiolabelled using $^{32}$P while the DNA/RNA elongation scaffold was labelled with the fluorescent dye Cy3 on the DNA template strand. The low mobility bands detected by the fluorescence signal in the upper panel of Fig. 3d correspond to bona fide TECs. The addition of $^{32}$P-RIP counteracts the Cy3-TEC signal in a

concentration-dependent manner. Vice versa, the addition of increasing amounts of Cy3-labelled nucleic acid scaffold leads to an increase in the signal corresponding to free $^{32}$P-RIP. In summary, the EMSAs support a competitive binding mode of RIP and the DNA/RNA nucleic acid scaffold. This is congruent with a model in which RIP can interfere with the DNA during loading, but not easily displace it once it is already loaded.

**Fig. 3 RIP occludes TFB and DNA binding to RNAP. a** Superimposition of the RNAP-RIP complex with the eukaryotic closed PIC (pdb 6gyk[20]). The archaeal RNA polymerase is shown in a grey surface representation and RIP as blue ribbon, in addition to eukaryotic TBP in light purple, TFIIB in red, and the promoter DNA in pink. The eukaryote-specific transcription factors that are not conserved in archaea including TFIIA were not included in the model. **b** Enlarged view of the the C-terminal tail of RIP mimicking the TFIIB linker in front of the RNAP rudder motif. **c** Superimposition of the RNAP-RIP complex with the open transcription initiation complex from *H. sapiens* (5iyb[26]) and *T. kodakarensis* (pdb 6kf9[63]). The same colour code used in **a** was applied with the euryarchaeal TFEα shown in light green. **d** RIP and DNA/RNA scaffold compete for RNAP binding. Electrophoretic mobility shift assay (EMSA) using Cy3-labelled DNA scaffold and $^{32}$P-labelled RIP (see SOURCE DATA). The fluorescence signal of the DNA is shown in the top panel and the radioisotope signal of RIP is detected by autoradiography and shown in the bottom panel. The (+) signs indicate 0.2 μM Saci RNA polymerase, and 0.4 μM RIP. Dose-response titration includes increasing concentrations of RIP (0.2, 1, 2 μM) or DNA/RNA scaffold (0.1, 0.5, 1 μM). The lower band visible at high DNA concentration (upper panel) is due to aspecific binding. The assay was successfully repeated four times. **e** Western blot showing the expression of RIP during late infection using polyclonal antibodies raised against RIP, with Alba serving as control (see SOURCE DATA). The western blot was repeated twice. **f** Superimposition of RNAP-RIP complex with the *B. taurus* RNAP-DNA-RNA transcription elongation complex (pdb 5oik[27]). The elongation factors Spt4/5 and Elf1 were not included in this model. The elongation factors Spt4/5 and Elf1 were not included in this model. The RNAP–RIP complex is coloured as in **a**, the eukaryotic DNA and RNA are shown in pink and cyan, respectively. **g** Enlarged view highlighting the clash of RIP with the NTS.

**RIP temporal expression pattern supports role in viroid maturation.** The role of RIP for virus fitness and function is still a matter of debate. As RIP directly interferes with nucleic acid binding and affects host and virus promoters alike, the early expression of RIP at high levels appears problematic. However, the global attenuation of transcription could benefit the virus by preventing transcription-dependent host defence mechanisms including the CRISPR IIIb system[28]. Alternatively, RIP could disengage RNAP from the actively transcribed viral genome aiding the DNA packaging into virus particles. The former would be associated with an expression pattern during early stage, and the latter during a late infection stage. To address this question, we analysed the temporal gene expression of RIP over a time course of 72 h. Exponentially growing *Acidianus* cells were infected with ATV and samples taken at regular intervals; RIP protein levels were detected by immunoblotting using a polyclonal antibody raised against recombinant RIP and compared to the expression levels of Alba, a chromatin protein that serves as control. In good agreement with its high toxicity, RIP could not be detected in *Acidianus* cell extracts during the early and middle stages of infection but was strongly upregulated at the end of the time course (72 h p.i.) just prior to cell lysis that occurs 96 h of post-infection (Fig. 3e). This suggests that RIP function is important during late infection, e.g., by dissociating RNAPs from the actively transcribed viral genome and thereby assisting maturation of the virion particle.

**The TFS4 cleavage factor paralog evolved into a RNAP inhibitor.** Unlike RIP, the TFS4 inhibitor is encoded by the host genome but only expressed in response to viral infection. Like RIP, TFS4 binds tightly to RNAP and efficiently represses transcription[16]. In order to investigate the structural basis for TFS4 inhibition and compare it to RIP's, we solved the cryo-EM structure of the RNAP–TFS4 complex at 2.6 Å resolution (Fig. 4a, Supplementary Fig. 6, and Table 1 for statistics). The map has been further refined by multibody refinement which allowed us to obtain a medium resolution map of the stalk at 3.8 Å, where all the known structural features were identified and correctly modelled inside the map (Supplementary Fig. 6g, h and Table 1). Archaeal TFS paralogues have a domain configuration akin to RPA12, RPB9 and RPC11 of RNAPI, II, and III, respectively, composed of two zinc-ribbon domains, ZR$^N$ and ZR$^C$, which are connected by a long linker (Fig. 4e). As proven by the cryo-EM map, the TFS4 ZR$^N$ interacts with RNAP between the upper jaw and the lobe (Fig. 4a, b). The N-terminal segment of the TFS4 linker forms two β-addition motifs (β3 and β4) by providing one antiparallel and one parallel strand to two β-sheets in the upper jaw of RNAP (Fig. 4b, e). The C-terminal segment (β5) of the linker, unexpectedly, packs on the TFS ZR$^C$ domain that interacts with the rim helices of the funnel (Fig. 4b, e). The TFS4 ZR$^C$

binds in the NTPs-entry funnel of RNAP in a manner that is related to its eukaryotic homologues[29–31], but without reaching through the pore into the active site (Supplementary Fig. 8e). The chemical nature of the interactions between the ZR$^N$ and ZR$^C$ domains with RNAP is also different (Supplementary Fig. 7b, c). The ZR$^N$-jaw/lobe interactions are dominated by a network of hydrophobic interactions, whereas the ZR$^C$-funnel interactions are facilitated by numerous hydrogen bonds and salt bridges which are not evenly distributed at the interface. The ZR$^C$ surface is abundant in positively charged residues (R57, R65, R70, K76-78, R79, and R82), most of which are not conserved in other TFS-related factors (Fig. 4c, e). This highly charged surface is balanced by interacting with two negatively charged patches formed by the lower alpha helix of the rim on one side (Rpo1′ S698, D702, and D705) and the upper jaw (Rpo1″ E301), which forms the internal wall of the funnel on the opposite side (Fig. 4b, d).

**Allosteric *modus operandi* of TFS4.** TFS4 overexpression induces retardation of cell growth in vivo, and TFS4 is a potent inhibitor of RNAP in vitro, increasing the $K_M$ for substrate NTP binding by ~50-fold and, like RIP, destabilises PICs and TECs[16]. Unlike RIP however, the binding sites of TFS4 with RNAP do not overlap with any of the RNAP interaction sites for the DNA or transcription initiation factors TBP, TFB, and TFE. Rather, TFS4 allosterically inhibits the RNA polymerase by inducing conformational changes driven by the displacement of the upper jaw, encompassing Rpo1″, Rpo5, 6, and 13, and the Rpo1′ clamp head (Fig. 5a). The jaw displacement allows TFS4 linker to bind to a surface that is otherwise occluded in the apo-RNAP, and is stabilised within the RNAP structure by the replacement of the two lost beta strands with two newly formed strands provided by TFS4 (Fig. 4 and enlargement in Fig. 5a). The superposition between the apo-bound and the TFS4-bound RNAP reveals a swinging movement of the jaw in unison with the clamp head by 6.3° concomitant with the splaying of the DNA-binding channel by 5 Å (Supplementary Movie 1). Considering that the width of the DNA double helix is 23.7 Å, the observed widening of the DNA-binding channel is likely to impair the close interactions of the DNA template with the RNAP, which provides the structural rationale for TFS4 destabilisation of RNAP-nucleic acid interactions during either initiation or elongation[16]. The Rpo1′ bridge helix spans across the DNA binding channel and is anchored on either side of it. The TFS4-induced opening stretches and bends the bridge helix by 3.8 Å and 2.2° leading to the loss of density between residues D808 to T813, likely due to the unwinding of the helix in the TFS4-bound RNAP (Fig. 5b). The ensemble of bridge helix and trigger loop plays a key role in the substrate nucleotide binding in the active site and its translocation cycle, i.e., the molecular mechanism underlying transcription elongation[32]. The TFS4 ZR$^C$ in the NTPs-entry pore clashes with the tip of the trigger loop resulting in its displacement, which is

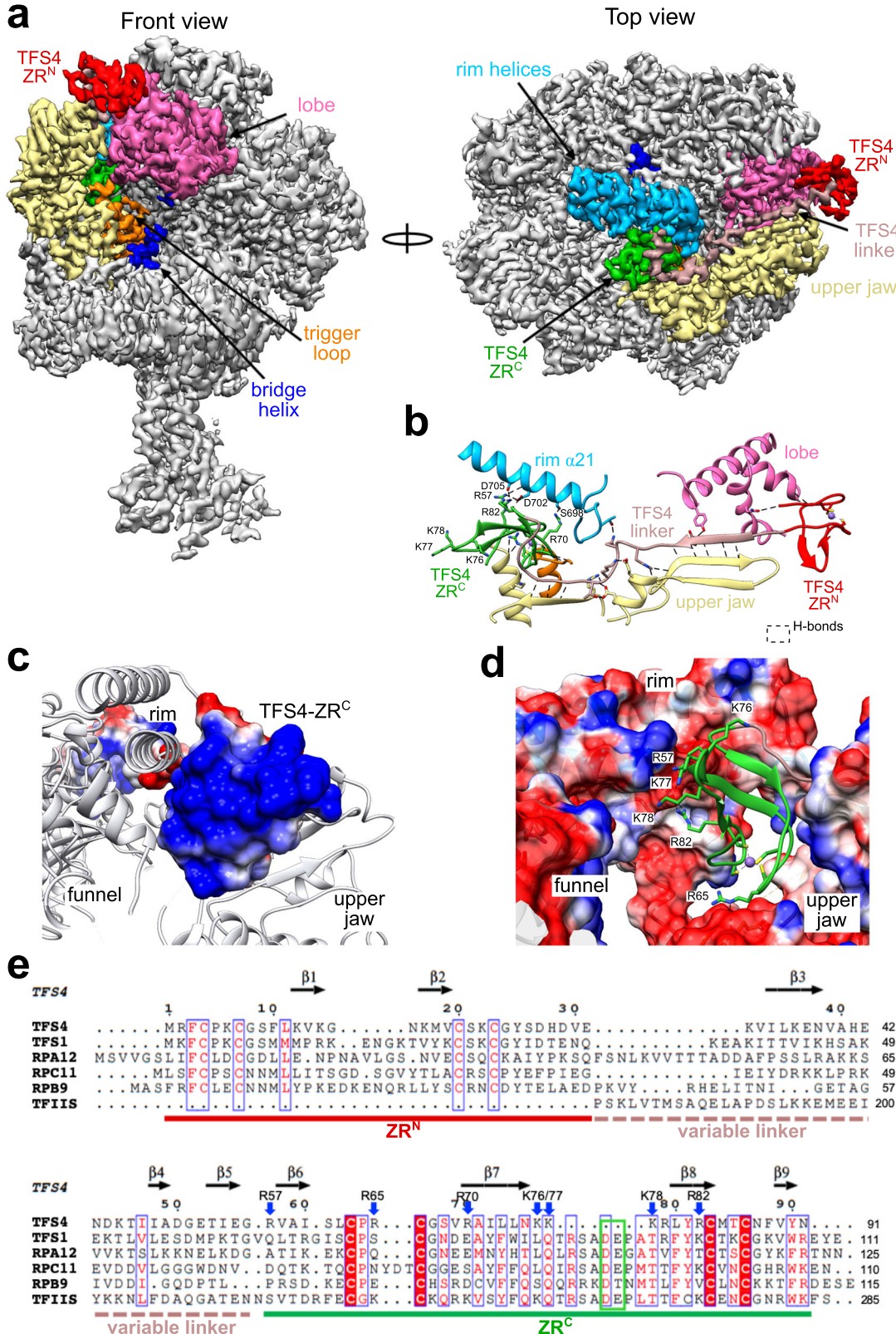

confirmed by the replacement of the RNAP trigger loop map with the TFS4 ZR$^C$ density and loss of structural information between residues R86 to E95 of Rpo1″ in the RNAP–TFS4 cryo-EM map (Fig. 5c).

In summary, the changes induced by TFS4 binding include the (i) widening of the DNA-binding channel, (ii) melting of the bridge helix, and (iii) displacement of the trigger loop. This three-pronged attack makes for a formidable intervention with the binding and catalytic mechanisms of RNAP.

## Discussion

As RNAPs are important therapeutic targets, the structural basis and mechanisms of their inhibition have been studied in great

**Fig. 4 Cryo-EM structure of the inhibited TFS4-RNAP complex. a** Cryo-EM map of the TFS4-bound RNA polymerase is shown in two orientations, where the domains of TFS4 and the RNAP motifs that interact with TFS4 are highlighted in different colours. The TFS4 ZR$^N$ and ZR$^C$ domaims are colour coded red and green, respectively, and the TFS4 linker in rosy brown. The RNAP rim helices are shown in sky blue, the upper jaw in yellow, lobe in dark pink, trigger loop in orange, bridge helix in blue. **b** The cartoon representation of TFS4 provides the details of the H-bonds (highlighted with dashed lines) interaction network with the RNA polymerase. All domains were coloured as on the cryo-EM map. **c, d** Electrostatic interactions facilitate the binding of the TFS4 ZRC in the RNAP funnel. The positively charged surface of the TFS4 ZR$^C$ domain that binds to the funnel rim helices is shown in **c**, and the negatively charged binding surface of the funnel is shown in **d** with the TFS4 ZRC shown as green ribbon with the zinc ion in purple. The residues directly involved in the binding to the RNAP as well as K76, K77, and K78 are shown in sticks with the H-bonds as dashed lines. **c, d** The positive charges are shown in blue, negative charges in red and non charged residues in white using a range of +5 to −5 kcal/(mol*e). **e** The structure-based sequence alignment of TFS1, TFS4, RPA12, RPB9, RPC11, and TFIIS was implemented with the secondary structure annotation for TFS4 on the top (Espript3 webserver) and domain organisation reported below the sequence alignment and coloured like panel **a**. The positively charged residues are indicated with a blue arrow on the top of the alighnment and the two catalytic carboxylate residues are highlighted within a green box.

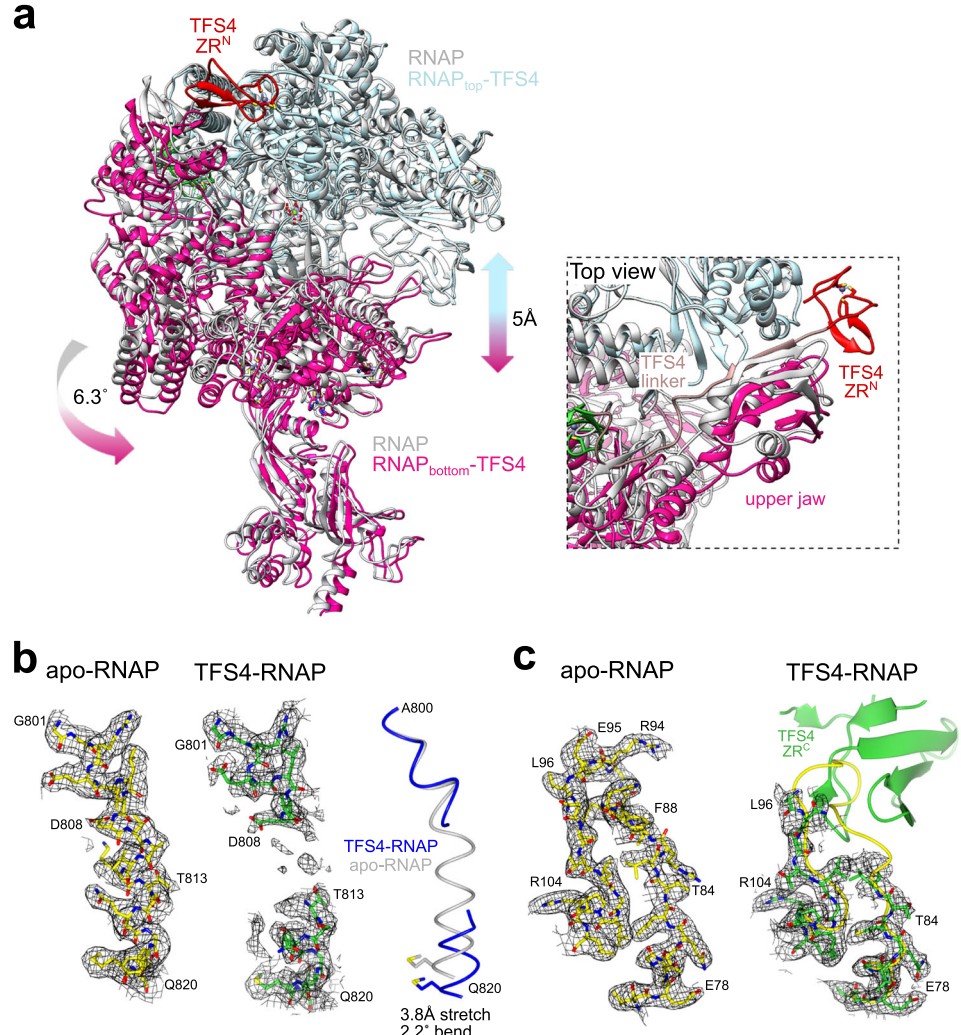

**Fig. 5 TFS4-induced allosteric inhibition of RNAP. a** Superimposition of the apo-RNAP and TFS4-bound RNAP illustrating largescale conformational changes resulting in a widening of the DNA-binding channel of 5 Å. Rpo2 subunits of apo-bound and TFS4-bound RNAP were superimposed in Chimera and the moving structural elements are highlighted in hot pink. In the enlargement a focus on the TFS4 linker position compared to the apo-RNAP showing how it replaces perfectly the beta sheets of the jaw. **b** Effect of the TFS4 binding on the bridge helix. From left the helix modelled inside the cryo-EM map for the apo-RNAP and the TFS4-bound one. The latter is shown with a high contour level to highlight the loss of density in the central section. On the right the superimposition of the bridge helix in the two structures is presented in worm style to appreciate the extend of the stretch and bending of the helix. **c** The TFS4 ZR$^C$ domain sterically clashes with the tip of the trigger loop, leading to loss of density, while the trigger loop is structurally well-resolved in the apo-RNAP.

detail in bacteria and eukaryotes—while nothing is known about archaeal RNAP. Arguing from first principles, inhibition can be achieved by (i) preventing direct interactions between RNAP and basal factors, nucleic acid template, or substrate NTP, or (ii) via allosteric mechanisms that alter the structure or conformation of RNAP in a way that abrogates catalysis. Our two novel structures of RNAP-inhibitor complexes both add new features and highlight common themes of intervening with RNAP function.

RIP is a small single-domain protein which binds to the inside of the RNAP clamp and rudder motifs, which is incompatible with the binding and function of the essential initiation factor TFB. The C-terminal tail of RIP and the B-linker strand and B-helix motifs of TFB (and TFIIB) interact with RNAP in an identical fashion (Fig. 3b)[33]. RIP forms a plug in the DNA-binding channel of RNAP, which stabilises and rigidifies the RNAP in the closed conformation. Such a binding mechanism interferes with the formation of the closed PIC, and it would prevent the loading of the DNA into the RNAP active site during the transition from the closed to the open PIC (Fig. 3a–c). The binding of RIP is furthermore incompatible with elongation by interfering with the stability of the DNA scaffold inside the DNA-binding channel (Fig. 3f). All of the observations above are in perfect agreement with published biochemical interaction analyses, which suggested a competitive inhibition mechanism of RIP. The initiation factor TFE induces an opening of the RNAP clamp[34], which stabilises the PIC and activates transcription[25]. Somewhat counterintuitively, complete TFE-containing PICs are more sensitive to RIP as compared to minimal PIC lacking TFE[7]. The structure of the RNAP–RIP complex rationalises this observation, as the TFE-induced opening the clamp[34] makes the RIP binding site more accessible. The occlusion of the DNA-binding channel is a reliable and direct mechanism of inhibition exploited by all domains of life. The bactoriaphage T7 Gp2 regulator binds within the DNA-binding channel of RNAP; it blocks the interaction between the downstream DNA and the RNAP β′ jaw domain and effectively inhibits the open complex formation of E. coli RNAP-σ70 (Fig. 6a)[35]. Among the eukaryotic transcription systems, the cellular negative regulator MAF1 specifically inhibits RNAPIII in response to stress and nutrient deficiency. Similarly to RIP, MAF1 binds inside the DNA-binding channel of RNAPIII and occludes the binding site of TFIIIB to the RNAP clamp and rudder motifs (Fig. 6b, c)[36]. What is singular about RIP and novel in the field is the unique ability of a viral protein to mimic the binding mode, and thereby prevent the binding of the universally conserved host basal initiation factor TFB.

Archaea, including Sulfolobales, encode functionally diversified paralogues of the transcript cleavage elongation factor TFS[16]. Sso TFS1 and TFS4 share the same domain organisation and a 28% sequence identity and both bind competitively to RNAP, yet while TFS1 stimulates elongation, TFS4 has evolved into a potent inhibitor of RNAP[16]. Although the structure of the RNAP-TFS1 complex has not been determined yet, it is very likely that the two conserved carboxylate residues in the TFS1 ZR$^C$ penetrate deep into the active site through the NTPs-entry funnel like TFIIS[11,16] as the molecular mechanisms of TFS1 and TFIIS are strictly conserved[12,13]. However, our structure shows that the TFS4 ZR$^C$ binds in the RNAP funnel, but not as deep and in a different orientation compared to TFIIS (Fig. 7b and Supplementary Fig. 8e, f). The TFS4 binding in the RNAP funnel is ensured by electrostatic interactions between the positively charged TFS4 ZR$^C$ domain and negatively charged patches on the rim helices, and the upper jaw of the RNAP as shown in the interface analysis of TFS4-RNAP complex (Fig. 4d and Supplementary Fig. 7b). Two acidic patches face the highly positively charged surface of the TFS4 ZR$^C$ domain (Fig. 4b, c, e). Site directed mutagenesis has indicated that three consecutive lysine residues (K76/77/78) in the TFS4 ZR$^C$ domain contributed to RNAP binding and inhibition[16], however, the lysine residues are surface exposed and do not make any contacts with RNAP in the RNAP-TFS4 structure. We surmise that the lysine residues enable an initial binding inside the negatively charged channel (Fig. 4c, d) that is followed by a rearrangement of the TFS4 ZR$^C$ domain inside the funnel resulting in the binding mode reflected in the RNAP-TFS4 structure. Interestingly, the propensity of the ZR$^C$ domains to flip in and out of the NTPs-entry channel is a well characterised mechanism of regulation during transcription elongation observed with RPA12 and RPC11 (RPC10 in human). Importantly, the inside funnel state of RPA12 and RPC10/11 is implicated in transcription termination of RNAPI and RNAPIII, respectively[31,37–39]. A superimposition of the apo-RNAP and RNAP–TFS4 complex reveals global as well as local changes of RNAP structure that account for the efficient inhibition, with an overall opening of the RNAP through the DNA-binding channel, and alteration of the bridge helix and trigger loop motifs in the active site (Figs. 5 and 7a).

The NTPs-entry funnel provides a crucial orifice to the RNAP active site that not only allows NTPs to enter, and the RNA 3′ terminus to exit the RNAP in backtracked TECs, but also serves as binding site for a plethora of transcription regulators[10]. These include negative regulators such as the bacterial Gfh1[8,40] and DksA[9,41] that have pleiotropic effects on transcription. During

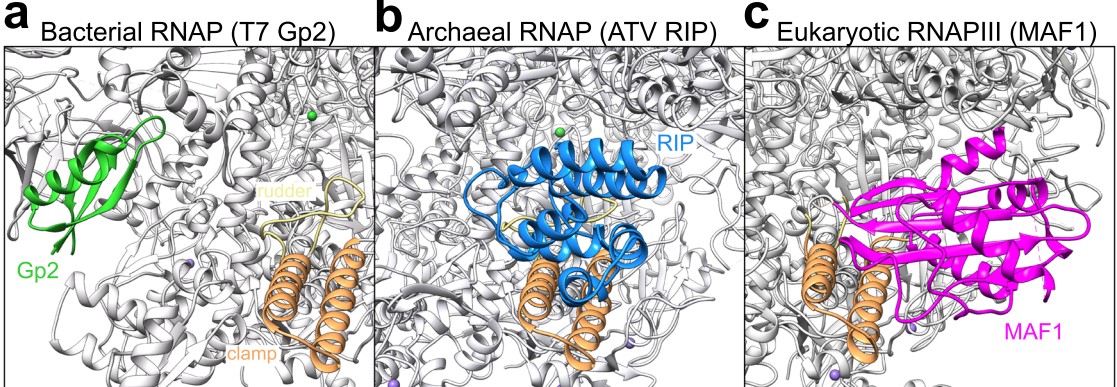

**Fig. 6 The DNA-binding channel is a common target for transcription inhibition in all domains of life.** The structures of the RNAP from **a** E. coli (pdb code 4lk0[6]), **b** S. acidocaldarius, and the **c** S. cerevisiae RNAPIII (pdb code 6tut[36]) are all shown in grey ribbons with the clamp and rudder highlighted in light orange and yellow, respectively. **a** The viral inhibitor T7 Gp2 is shown in green, **b** RIP from ATV in blue, and **c** the cellular negative regulator MAF1 in magenta.

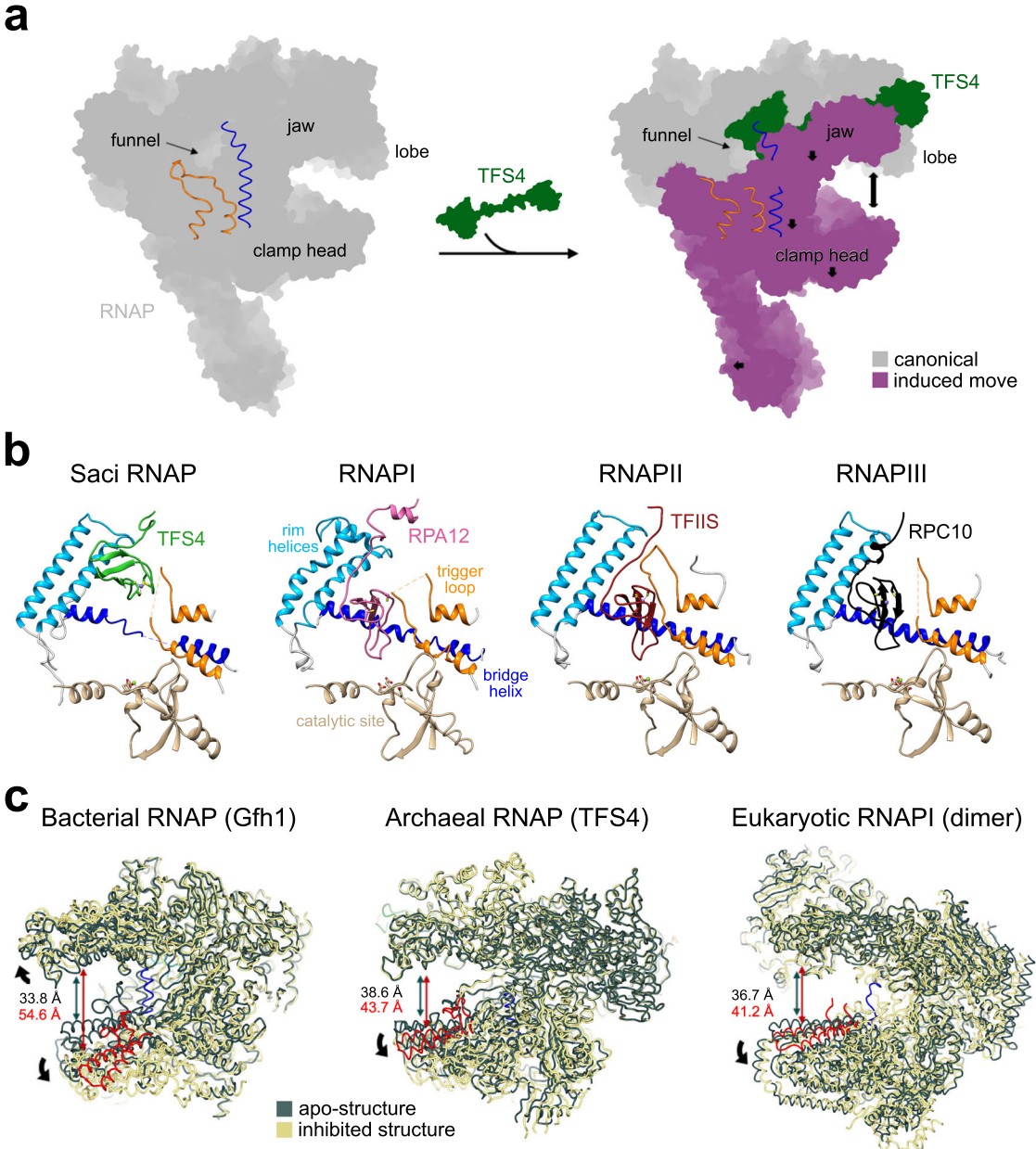

**Fig. 7 Commonalities into allosterically exploiting the RNAP structural flexibility. a** Schematic representation of the conformational changes induced by TFS4 binding. The RNAP is shown in grey, and the part of the structure subjected to conformational change is highlighted in purple with the movement direction indicated by arrows. Main features of RNAP including the secondary channel, jaw, lobe and clamp head, as well as bridge helix and trigger loop are indicated. **b** Comparison of the archaeal Saci RNAP with eukaryotic RNAPI, II and III and the locations of the ZR[C] domains of subunits TFS4, RPA12, TFIIS, and RPC10 (human homologous of RPC11) (pdb codes 6rqh, 5xon, and 7ae3, respectively for the eukaryotic factors[69,70,31]), relative to the rim helices and the active site motifs (aspartate triad, bridge helix and trigger loop). Only the TFS4 ZR[C] domain is located at the rim of the NTPs-entry funnel and clashes with the trigger loop, while all other ZR[C] domains reach towards the catalytic magnesium ion (green sphere). **c** The structural changes associated with the inhibition of the archaeal, bacterial and eukaryotic RNAPI by TFS4, Gfh1, and RNAPI dimerisation, respectively, involve a widening of the DNA binding-channel. The structures of the free, or apo forms of RNAP are shown in dark teal, and the inhibited conformations of the RNAP-TFS4, RNAP-Gfh1 and RNAPI dimers are shown in yellow, respectively. The RNAP clamp helices are highlighted in red and the bridge helix distortions in blue.

the stringent response induced by amino acids starvation, DksA in conjunction with ppGpp is thought to alter and destibilise conformational transitions en route to the open complex of stringent promoters. The binding of ppGpp, even in the absence of DksA, restricts the conformational flexibility and thereby inhibits RNAP[42]. Gfh1 inhibits all catalytic activities of RNAP enhancing pausing[40,43,44]. Like TFS1 and TFS4 in archaea, the bacterial GreA, GreB, Gfh1, and DksA factors compete with each other for RNAP binding, and changes in their relative expression

level is likely to be an integral part of their regulation. In the case of TFS1 and 4, the former is constitutively expressed and engaged with RNAP, while the latter is exclusively expressed in response to viral infection[17]. TFS4 and Gfh1 are not homologous but share intriguing functional properties that likely result from convergent evolution. Both bind to the RNAP funnel and allosterically inhibit RNAP which results in the widening of the DNA binding channel and—albeit to a different degree—a distortion of the bridge helix and trigger loop in the active site (Fig. 7c).

What is the origin of the conformational dynamics of DPBB RNAP in all domains of life? The NTP translocation cycle of RNAP involves tightly coordinated conformational changes as it cycles through post-translocated and pre-translocated, and of NTP pre-insertion and insertion states that have been captured by high-resolution X-ray structures of the bacterial RNAP[45,46]. As the archaeal RNAP progresses through the transcription cycle, the flexible RNAP clamp adopts distinct conformational states, which alter the width of the DNA-binding channel[34]. Elongation is a discontinuous process interrupted by pausing and termination, which involves inhibited states. Recently, it has been shown that the bacterial RNAP in the context of an RNA hairpin in elemental pause—which is likely preceeding termination—has a widened DNA-binding channel, which is reminiscent of the archaeal TFS4-RNAP complex[47]. This is different from a closed RNAP clamp that is characteristic for processive elongation complexes[45,46]. Equally, in the inactive dimeric form of RNAPI, the two macromolecules face each other with the DNA-binding channel of one molecule interacting with the stalk of the other[29]. In this conformation, the RNAPI jaw and lobe modules have moved outwards and the DNA-binding channel is widened by 5 Å[48] similarly to the inhibited state of the archaeal TFS4-RNAP complex (Fig. 7c). Finally, eukaryotic RNAPI, II and III include RPA12, RPB9 and RPC10/11 subunits, respectively, that are paralogs of TFIIS but stably incorporated into the RNAP rather than reversibly associated with it (Supplementary Fig. 8a–d).

We posit that the allosteric changes induced by TFS4 are related to several inhibited states of DPBB RNAP, including transcription effectors (like Gfh1), oligomerisation equilibria (RNAPI) and more general inhibited states of RNAPs associated with pausing and termination. In the structurally well characterised cases discussed above, the DNA-binding channel is widened and the interactions with the template DNA weakened. The common denominator of inhibition is the ability to lock them in a specific conformation. In a likely scenario, RNAP subunits and factors have evolved vertically by gene duplication and speciation (e.g., TFIIS, TFS1, and −4, RPA12, RPB9, and RPC11/10) or by convergent evolution (e.g., GreA/B and Gfh1 relative to TFS1) to exploit the inherent conformational flexibility of RNAP to modulate and fine tune transcription.

The biological function of RIP for the ATV virus is still a matter of debate, but its tight regulation and strong induction during the very late stages of lytic infection supports a role in virus particle assembly, possibly by dissociating transcribing RNAP from viral genomes. The strong and stable interactions between RIP and the RNAP clamp and rudder make RIP a lethal protein, and this absolute mechanism to shut down RNAP is apparently the preferred option for viruses and bacteriophages like ATV and T7[6].

In comparison, host-encoded factors such as TFS4 and Gfh1 seem to opt for a more versatile approach by allosterically interfering with RNAP function. This provides an opportunity for the cell to temporarily pause transcription, mount additional defence mechanisms, and eventually reactivate the gene expression programme once the favourable conditions have returned. This is the case for the well-studied Gfh1 which adopts an inactive conformation during normal growth conditions, and undergoes a reversible conformational change upon acidification of the medium[22]. The plasmid-driven expression of TFS4 stops cell growth similar to virus infection, but it is not known yet whether TFS4 is the sole agent to trigger the host growth retardation in response to infection[16]. The induction of a quiescent state is an important preamble for persistence of many pathogenic bacteriophages[49], hence, we hypothesise that TFS4 may play a similar role in archaea.

The detailed structural bases and the mechanisms, which underlie the inhibition of archaeal DPBB RNAP described in this manuscript, have the potential to aid the design of novel drugs targeting the RNAPs of bacterial pathogens and RNAPI in cancer therapy[50]. These include small effector proteins that bind tightly to RNAP with high specificity, like RIP, and either deny access to the DNA binding channel or essential transcription factors. Such effectors could be eventually delivered to their eukaryotic or bacterial targets by recombinant viruses or bacteriophages, respectively[51]. Moreover, agents that bind to RNAP in a polydentate fashion like TFS4 and prevent conformational changes that are critical for catalysis are possibly more refractive to elicit the fast emergence of resistance mutations.

## Methods

**Protein expression and purification**. Saci RNA polymerase, 6×His tagged at the C-terminus of Rpo8, was expressed in Saci strain MW001 and purified according to established protocols[16]. Cells were resuspended in buffer 50 mM Tris pH 8.0, 200 mM NaCl, 25 mM Imidazole, 1 mM DTT, 100 µM ZnSO$_4$, 5% glycerol supplemented with one tablet of protease inhibitors (cOmplete Tablets, Roche), 1 µl/ml DNase I, and 0.5 µl/ml RNase A. Cell suspension was sonicated for 30 min at 70% amplitude in pulse mode on ice, followed by centrifugation at 60,000 × $g$ at 10 °C for 30 min. Cell extract was first loaded onto Ni-column (the lysis buffer was supplemented with 250 mM Imidazole for the elution), followed by Heparin purification (the sample was loaded at 50 mM NaCl buffer and eluted in gradient up to 1 M NaCl). Fractions containing DNA-free RNAP were pooled, concentrated in Amicon 100 kDa cutoff (Millipore), and loaded onto Superose 6 10/300 column (GE Life Science).

ATV RIP (ORF145), tagged at the N-terminus with a cleavable 6×His sequence, was over-expressed in BL21 pLyS by induction with 1 mM IPTG. After lysis by sonication (6 min in pulse mode at 70% amplitude on ice) in buffer 50 mM Tris pH 8.0, 200 mM NaCl, 25 mM Imidazole, supplemented with one tablet of protease inhibitors (cOmplete Tablets, Roche, the cell extract went through a precipitation heat step at 60 °C for 20 min, followed by centrifugation at 60,000 × $g$ at 10 °C. The cleared cell extract was purified in Ni-column and the tag cleaved in dialysis at room temperature (rt) with TEV protease. The sample was further purified in Ni-column to remove the tag[7].

Sso TFS4 (without tag) was expressed in BL21 pLysS by induction with 1 mM IPTG. Cells were resuspended in buffer 50 mM Tris pH 8.0, 50 mM NaCl, 1 mM DTT, 100 µM ZnSO$_4$, 5% glycerol supplemented with 1 tablet of protease inhibitors (cOmplete Tablets, Roche), 1 µl/ml DNase I, and 0.5 µl/ml RNase A, and sonicated for 6 min in pulse mode at 70% amplitude on ice. The cell lysate was incubated at 65 °C for 20 min to promote bacterial proteins precipitation, and the cell extract cleared by centrifugation at 60,000 × $g$ at 10 °C. The sample was purified with HiTrap Q column (GE Life Science) using 1 M NaCl for the elution in gradient, followed by Superose 12 10/300 column (GE Life Science)[16].

**Complex assembly and cryo-electron microscopy data collection**

*Apo-RNAP*. The RNA polymerase was crosslinked at 0.15 mg/ml in 200 µl of a buffer containing 20 mM Hepes pH 7.0, 200 mM NaCl, 5 mM MgCl$_2$, 100 µM ZnSO$_4$, 10% glycerol, 5 mM DTT, with 2 mM bis(sulfosuccinimidyl)suberate (BS$^3$) for 5 min at 65 °C. The reaction was quenched adding 150 mM NH$_4$HCO$_3$ at rt for 20 min. The sample was then diluted ten times in the same buffer without glycerol, filtered with a 0.22 µm filter and concentrated up to 0.4 mg/ml in a concentrator with a cutoff of 100 kDa. Sample quality was firstly assessed by negative staining, then 3 µl of sample at 0.06 mg/ml was spotted on a UltrAuFoil holey grid 300 mesh R1.2/1.3 (Quantifoil, Germany) covered with graphene oxide according to a protocol described by Cheng K. and co-workers[52], and vitrified by plunging in liquid ethane using Vitrobot Mark IV (Thermo Fisher Scientific, USA) at 4 °C and 94% humidity. Data were collected at eBIC National facility (Diamond Light Source, UK) using a Titan Krios microscope (Thermo Fisher Scientific, USA) operated at 300 keV and equipped with a BioQuantum energy filter (Gatan, USA). The images were collected with a post-GIF K3 direct electron detector (Gatan, USA) operated in super resolution mode, at a nominal magnification of 81,000, corresponding to a pixel size of 1.085 Å. The dose rate was set to 21 e$^-$ per pixel per second, and a total dose of 44.46 e/Å$^2$ was fractionated over 40 frames. An energy slit with a 20 eV width was used during data collection. Data were collected using EPU software (Thermo Fisher Scientific, USA) with a nominal defocus range −1.0 to −2.5 µm. During the data collection microscope stage was tilted to −30° to overcome preferred orientations.

*RNAP/RIP complex*. Prior the crosslinking procedure already described, the RNA polymerase was incubated with a 10-fold excess of RIP for 5 min at 65 °C. Grids covered with graphene oxide were prepared according to the protocol described above, and data were collected in ISMB Birkbeck EM facility using a Titan Krios microscope operated at 300 keV and equipped with a BioQuantum energy filter. The images were collected with a post-GIF K2 Summit direct electron detector (Gatan, USA) operated in counting mode, at a nominal magnification of 130,000 corresponding to a pixel size of 1.047 Å. The dose rate was set to 5.84 e$^-$ per pixel

per second, and a total dose of 45.5 e/Å$^2$ was fractionated over 45 frames. An energy slit with a 20 eV width was used during data collection. Data were collected using EPU software with a nominal defocus range −1 to −2.5 μm. During the data collection microscope stage was tilted to −30°.

*RNAP/TFS4 complex.* The RNA polymerase was incubated and crosslinked as described above in presence of a 20 molar fold excess of TFS4. The sample at 0.4 mg/ml of concentration was applied twice on a UltrAuFoil holey grid 300 mesh R1.2/1.3 and vitrified by plunging in liquid ethane using Vitrobot Mark IV at 4 °C and 94% humidity. Data were collected at eBIC National facility using a Titan Krios operated at 300 keV and equipped with a BioQuantum energy. The images were collected with a post-GIF K3 direct electron detector (Gatan, USA) operated in super resolution mode, at a nominal magnification of 81,000, corresponding to a pixel size of 1.085 Å. The dose rate was set to 21.221 e$^−$ per pixel per second, and a total dose of 45.2 e$^−$/Å$^2$ was fractionated over 40 frames. An energy slit with a 20 eV width was used during data collection. Data were collected using EPU software with a nominal defocus range −1 to −2.5 μm.

**Cryo-electron microscopy data processing**

*Apo-RNAP.* The dataset of 1676 movie stacks was aligned, summed and 2× binned using MotionCor2[53], followed by CTF estimation using GCTF[54]. Relion 3.0 software[55] was used for template-free particle picking (Laplacian method) and all consequent image processing for this sample. Initially 1,286,432 particles were extracted and downscaled to a/pix of 4.5 Å. After multiple cycles of 2D and 3D classifications 423,157 best particles were selected and rescaled to the pixel size of 1.085 Å. The selected subset of particles was then refined and, after post-processing, subjected to four cycles of CTF refinement to correct for the effect of stage tilt used during data collection on the initial CTF estimation. Three cycles of CTF refinement were sufficient for the correction, and the fourth cycle didn't provide any further improvement. The final cycle of 3D refinement and post-processing resulted a map with resolution of 2.88 Å as estimated using gold standard Fourier Shell Correlation (FSC) with a 0.143 threshold (Supplementary Fig. 1 and Table 1).

*RNAP/RIP complex.* The 2130 movie stacks were aligned and summed using MotionCor2 followed by CTF estimation in GCTF. Particles were picked using Relion 3.0 reference-based method. These 600,640 particles extracted from the micrographs were subjected to multiple rounds of 2D and 3D classifications using cryoSPARC[56]. Multiple approaches were applied to identify the apo-RNAP but no additional species were found likely due to the relatively low number of particles available for the search. The best 151,237 particles were rescaled to the original pixel size, and image processing continued using Relion 3.0. Particles were refined and subjected to three cycles of CTF refinement. After a final 3D refinement and post processing step the resolution of 3.27 Å was estimated using gold standard FSC with 0.143 threshold (Supplementary Fig. 5 and Table 1).

*RNAP/TFS4 complex.* The dataset of 1760 movie stacks was motion corrected and analysed with Relion 3.0 following the same routine described above for the apo-RNA polymerase sample. After multiple rounds of 2D and 3D classifications, best 505,758 particles were selected out of 1,161,535, and subjected to two cycles of CTF refinement followed by particle polishing. Although the large number of particles, we did not identify the apo-RNAP or any other intermediate conformational species, using either Relion 3.0 and Cryosparc. The last 3D refinement and post-processing steps provided a map with an averaged resolution of 2.59 Å as estimated using gold standard FSC with 0.143 threshold. Analysis of the Euler angles distribution for this map revealed a clear cluster of preferred orientations. Although it didn't seem to affect map quality, we decided to prune the refined particles and repeat refinement and post-processing. This step was carried out using an in-house script considering the tilt and rotation angles of the last refinement, and pruning the particles based on the CtfFigureOfMerit metadata up to 350,000 particles. The pruning provided a batch of 350,682 particles, which gave a map with resolution of 2.61 Å at the FSC of 0.143 (Supplementary Fig. 6 and Table 1). The local resolution of TFS4 map shown in Supplementary Fig. 6d resulted to be perfectly compatible with the local resolution of the RNA polymerase in the same area suggesting that the apomediate or intermediate species, if present, represent all together less than the 5% of the dataset, which might explain the resulting single class from the 3D classification step. To improve map quality of the stalk which is intrinsically a flexible arm protruding from the main body, we performed a multi-body refinement using the RNAP main domain and the stalk as moving bodies. To do that, we generated two masks with soft edges for the large globular main domain and the stalk, respectively, to which we applied 20° of width for the rotation priors and five pixels for the translation between the two bodies, a setup suggested for highly flexible bodies according to the protocol released by Nakane T and Sheres SHW[57]. The multibody refinement was the only successful approach found improving the stalk resolution up to 3.75 Å (Supplementary Fig. 6g, h and Table 1).

**Model building and refinement.** Local resolution was assessed using Relion 3.0 after post-processing, whereas map sharpening and model refinement were carried out in Phenix v1.15.2 and 1.19.2[58], including rounds of manual editing and refinement in Coot v0.8.9.1[59] (Table 1). Models of all RNAP subunits, RIP and

TFS4 were prepared using Modeller[34], then the RNAP complex was assembled using the homologous RNA polymerase from *Sulfolobus shibatae* at 3.2 Å (pdb 4ayb[19]) and refined against the cryo-EM map of RNAP/TFS4 at 2.61 Å. The structure obtained was used to refine the model against the maps of the apo-bound and RIP-bound RNA polymerase. Following the RNAP refinement, the initial models of RIP and TFS4, obtained using Modeller, were placed inside the extra-densities and refined accordingly. The C-terminal tail of RIP and TFS4 linker, not predicted in the initial model, were built manually in Coot following map density.

**Interfaces data analyses and sequence alignments.** The binding interfaces between the RNA polymerase and RIP or TFS4 were analysed using two different programmes for consistency, PISA and LigPlot+[60,61]. As setup in LigPlot+ we used for the H-bonds and salt bridges a maximum of 3.5 Å, for the hydrophobic interactions we used the maximum values of 3.5 Å.

Structure-based alignments of all Saci RNAP subunits have been carried out with the Match-Align tool in Chimera[62] after superimposition of the *S. acidocaldarius* apo-RNAP with *S. shibatae* (4ayb[19]), *S. solfataricus* (3hkz[18]), *T. kodakarensis* (6kf9[63]) and *S. cerevisiae* (5vvs[64]) followed by manual editing of the flexible loops. ATV RIP and P131/ORF131, and SMV-1 ORF122 and ORF114 have been aligned using PROMALS3D[65]. Alignment images throughout the text were prepared in Espript3[66].

**ATV infection and biomass sampling.** Acidianus convivator strain AA9 was grown at 76 °C in 1× medium including sulphur[67]. The cell density was monitored with a Shimadzu spectrophotometer (OD600) and the cell number determined with a Thoma counting chamber at different times points during the growth. Cells from 500 ml of exponentially growing A. convivator were collected at OD600 = 0.08 by centrifugation at 3500 rpm for 30 min at 4 °C, and the pellet was suspended in 500 μl of 1× medium. A ATV virus preparation of 150 μl[68] (titre of 1010 virions/ml) was added to the cell suspension, the mixture was incubated at 80 °C for 1 h and diluted with 500 ml of 1× medium including sulphur. Infected cells were grown at 76 °C and samples taken at t = 0, 6, 12, 18, 48 and 72 h post-infection.

**Immunodetection.** Cell pellets were sonicated (20% amplitude, 10 s pulse mode for 1 h) and protein content was measured using the Qubit system (Invitrogen). For each sample, the equivalent of 9 μg of total protein was loaded on a 14% Tris-Tricine SDS-PAGE and blotted on a 0.2 μm PVDF membrane. As primary antibodies, we used a 1:1000 dilution of a rabbit polyclonal anti-RIP (Davids Biotechnology), and a 1:3000 dilution of a sheep polyclonal anti-Sso Alba (kindly provided by Malcom White, University of St. Andrews, UK), the second target used as additional internal control to assess protein content of the samples. To visualise the two protein targets in different colours we used two different secondary antibodies, the donkey anti-rabbit IgG was conjugated with Dylight 680 and used at 1:10,000 dilution (A120-208D6, Bethyl Laboratories, Cambridge, UK), the donkey anti-sheep IgG was conjugated with Dylight 488 and used at 1:2000 dilution (A130-100D2, Bethyl Laboratories, Cambridge, UK). Fluorescent detection was carried out on Typhoon FLA 9500 scanner (GE Healthcare) for Alexa-488 and Alexa-680.

**EMSA experiments.** The DNA:RNA scaffold was prepared incubating the DNA template strand labelled with Cy3 (Cy3-TS83) with RNA (RNA14) at 76 °C for 5 min, then we added the non-template strand (NTS83) and we incubated for further 5 min at 76 °C. RIP was cloned into the pKA-vector carrying the phosphorylation site at the N-terminus. It was expressed, purified and labelled with $^{32}$P according to published protocol[7]. In order to be able to use detectable signals of both fluorescent scaffold and radioactive RIP, $^{32}$P-RIP was mixed with cold RIP in to 3:1 ratio. We split the competition assay in two halves. In the first half assay we pre-incubated the scaffold (100 nM) with Saci RNAP (200 nM) in 15 μl of buffer 10 mM MOPS pH 6.5, 10 mM MgCl$_2$, 200 mM NaCl, 10% glycerol, 0.067 mg/ml BSA, and 0.1 mg/ml heparin at 65 °C for 5 min, then we added increasing amount of $^{32}$P-RIP (0.2–2 μM) and we incubated again at 65 °C for 5 min. The competition reactions of the second half assay were prepared by pre-incubating $^{32}$P-RIP (400 nM) with 200 nM Saci RNAP in 15 μl of the same buffer at 65 °C for 5 min, then adding 0.1–1 μM scaffold and incubating for further 5 min at 65 °C. Samples were resolved on a 6% native PAGE and detected using Typhoon (GE).

Oligo sequences are reported in Supplementary table 1.

**Reporting summary.** Further information on research design is available in the Nature Research Reporting Summary linked to this article.

## Data availability

Structural data generated in the current study are available on the RCSB Protein Data Bank and the corresponding EM maps deposited on the Electron Microscopy Data Bank. PDB accession codes: apo-RNA polymerase 7ok0, RNAP/RIP complex 7oq4, RNAP/TFS4 7oqy. EMDB accession codes: RNA polymerase EMD-12960, RNAP/RIP complex EMD-13026, RNAP/TFS4 EMD-13034. Source data are provided with this paper.

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

## Acknowledgements

We would like to thank Diamond Light Source for access to the cryo-EM facilities at the UK National electron Bio-Imaging Centre (eBIC, proposal EM20287) funded by the Wellcome Trust, the Medical Research Council UK and the Biotechnology and Biological Sciences Research Council. Cryo-EM data for this investigation were also collected at the ISMB EM facility at Birkbeck College, University of London with financial support from Wellcome (202679/Z/16/Z and 206166/Z/17/Z). We would like to thank Dr. David Houldershaw for IT support and Giulia Zanetti for the script writing. We are very grateful to Jerome Gouge, Anthony Roberts and Christoph Müller for discussions, comments and suggestions. Research in the RNAP laboratory at UCL is funded by a Wellcome Investigator Award in Science to FW (WT 207446/Z/17/Z) with the title Mechanisms and Regulation of RNAP transcription.

## Author contributions

S.P., N.L., L.M.D. and A.C. worked on cryo-EM data, T.F. and C.S. prepared proteins and carried out cross-linking and Western blotting, S.L.S. and D.P. carried out virus infection experiments, D.M. contributed to the identification of Rpo8 mismatch, S.P. and F.W. wrote the manuscript, and F.W. conceived and planned the project.

## Competing interests

The authors declare no competing interests.
