## [Peer Review File · Nature Communications]

Structural basis of RNA polymerase inhibition by viral and host factorsREVIEWER COMMENTS

Reviewer #1 (Remarks to the Author):

In this manuscript, Pilotto et al. report cryo-EM structures of *Sulfolobus acidocaldarius* RNA polymerase (RNAP) in its apo-form as well as in complex with two different proteins that inhibit its activity, RIP and TFS4. While RIP is a viral protein expressed by ATV, TFS4 is a cellular protein expressed by the host. The same group has previously characterized RIP biochemically (reference 22), and the work presented here provides very compelling structural data on the mechanism of this viral RNAP inhibitor. This data is in excellent agreement with their previous hypotheses, and provides evolutionary insights into how this protein repurposes a capsid fold for the inhibition of transcription by sterically competing with the DNA and with initiation factors. In addition, the authors show that the expression of RIP is specific for late stages of infection, which provides clues to its role in the viral life cycle. The structure of TFS4 bound to RNAP shows that this factor is structurally related to transcription factors or RNAP subunits that promote transcription elongation and binds to a similar position as these. In contrast to these, however, it disrupts critical elements required for catalysis in the RNAP active site and thereby inhibits transcription. In addition, its binding leads to opening of the DNA binding channel, which is predicted to impair nucleic acid binding. Finally, the authors compare the mechanisms of RIP and TFS4 to other known inhibition mechanisms of multi-subunit RNAPs, which emphasizes common principles of transcription inhibition.

The results presented here provide detailed molecular mechanisms for the inhibition of archaeal RNAP by RIP and TFS4. These data are important and novel findings that are not only of interest to those studying archaea, but also for the broad field of transcription and its regulation. The conclusions are well supported by the data, and the structure analysis by single-particle cryo-EM is state-of-the-art. I have no technical concerns, and no further experiments are needed to support the claims. In conclusion, I think this manuscript should be published after addressing some open questions and comments outlined below, which mainly concern the presentation and discussion of the data.

Hauke Hillen

Major points:

- The authors describe a novel zinc binding site in Rpo1', which they term Zn3. In Figure S3a it appears that the zinc ion is coordinated by only three residues (Cys, Cys, His). A coordination number of 3 is extremely rare for zinc in protein structures (PMID: 24059258). Is there a fourth coordination partner in vicinity? If not, is there any additional evidence in the literature that this motif indeed binds zinc?

- Lines 111 – 118: In this paragraph, the authors describe the structure of apo-RNAP (determined at 2.9 Å) and that the high quality of the cryo-EM reconstruction allowed modeling of various structural features. However, in some of the referenced figures (S3), the 2.6 Å cryo-EM density from the subsequently described RNAP-TFS4 dataset is shown to emphasize these features. Which density is shown in Figure 1B and 1C? The authors should clearly state this to avoid confusion.

- Lines 115 – 118: This sentence seems confusing, because it has not been stated how the

initial model of Rpo8 was obtained, and it is therefore not clear what was rebuilt. I would suggest to include a half-sentence along the lines of “Compared to previous archaeal RNAP structures, ...” to clarify this.

- Lines 181 – 187: The sentence starting with “ATV has a broad host range...” should be more specific: I assume it is not the virus that inhibits *M.jannaschii* RNAP, but purified RIP. Figure S6d shows that several of the critical residues appear to be conserved in *S.cerevisiae* Pol II. Is it known whether RIP also inhibits eukaryotic RNAPs? Are homologues of RIP encoded by viruses that infect eukaryotes?

- Figure 3A: The description of what is shown is not entirely clear: Are the polymerase model and RIP from the structure described in this paper and the DNA, TFIIIB and TBP from 6gyk? Or was RIP from the current structure simply overlaid with 6gyk? It also appears as if TFIIA (chains U and V) were omitted, which should be stated in the legend. The same applies to panel C and F.

- The authors show that RIP occupies the same binding site as the DNA and TFIIIB-like factors and that adopts a “similar binding mode to the RNAP rudder” (line 210). Does this refer to conserved interactions on the amino acid level with the RNAP or similar structural motifs, or does it merely occupy the same binding site?

- Lines 226 – 228: The authors conclude that their data support a competitive binding mode of RIP and the DNA/RNA nucleic acid scaffold, which is also supported by the structure. However, previous data by the authors (reference 10) indicated that RIP and the nucleic acid do not compete with each other for binding to RNAP. Is there an obvious explanation for these results?

- Figure 3E and lines 235-262: The experiment showing that RIP is apparently exclusively expressed during late stages of infection is very striking. Is it known what other ATV genes follow a similar expression pattern? Is this the stage when capsid proteins are also expressed? Are there any indications that RIP could act in concert with other viral factors?

- Figure 4: I had a hard time correlating the description of results (paragraph lines 264 – 307) with this figure, and I think it could be more intuitively understood with a few rearrangements. First, I found it difficult to decipher which parts belong to TFS4 and which parts belong to RNAP in panel A and C. I think the figure would benefit from a schematic depiction of TFS4 (for example as bar representing the primary sequence), with domain annotation and colored accordingly as in panels A and C. For panel A, may I suggest the authors depict structural models instead of cryo-EM density as surface? I find the latter adds relatively little information content, but makes it difficult to see details (this also holds true for Figure 2C, but there it is clearer how and where RIP binds). One possibility would be to show the RNAP model as surface rendering and slightly transparent, and only TF4S and the elements of RNAP that interact with it as cartoon or surface and colored (for example as in another paper by this group, ref 22 Figure 7a). In combination with the above-mentioned schematic of TF4S, I believe this would make the figure much more intuitive. Regions shown enlarged in panels B and C could then possibly also be indicated by small boxes around them. Moreover, in panel C, some residues in the jaw and TFS4 are emphasized as sticks, but not labeled, and it is thus not clear where for example the important lysine cluster (K76/77/78) is located. Finally, I would suggest the authors consider swapping Figure 4D with the structural comparison in Figure S7a-d. This could then be referenced at the end of the sentence in line 288.

- This group has previously shown that a lysine cluster in TF4S (K76/77/78) is critical for its function in RNAP inhibition (reference 22). Does the RNAP-TF4S structure provide direct evidence for the role of these residues? In Figure 4B and C, they appear to be depicted no interactions are obvious. In lines 298 – 302, the authors speculate that these residues may play a role during initial binding of TFS4 to RNAP, and that TFS4 undergoes a remodeling during binding to RNAP. Is this somehow inferred from the structural data presented here, or is it a hypothesis? In the latter case, I think it should rather be moved to the discussion.

- Organizational issue: In lines 288 – 289 the authors mention that TFS4 clashes with the trigger loop without occluding the pore, but this is only explained in the next section (from lines 321 on) and no reference is given to Figure 5 where it is depicted. Therefore, this statement stands isolated here. Similarly, the section in lines 302 – 307 discusses the similarities and differences between TFS4 and other rim-binding factors and emphasizes that TFS4 is the first example of a zinc ribbon domain that clashes with the trigger loop and induces conformational changes. However, this is only described in the section that follows. I think both the first statement (lines 288 - 289), and the concluding remarks (302 – 307) would be more appropriately placed in the following section (lines 321 and onward). This way, the first section on TF4S would focus on how the factor binds to RNAP, while the second section would describe its mechanism of inhibition.

- Lines 324-326: The authors state that the binding sites of TFS4 do not overlap with those of initiation or elongation factors. However, from my understanding and the depictions in Figure S7a-d, the binding site of TFS4 does overlap with that of RPA12 / TFIIS / RPC10 in other transcription systems and likely with that of TFS1 in the archaeal system (reference 22). Could the authors clarify this?

- Since the authors describe in the methods that specific experimental approaches were employed to overcome particle orientation bias (graphene oxide, tilting of the stage), I would strongly encourage them to include angular distribution plots in Figures S1, S4 and S5 to demonstrate the outcome of these attempts. In addition, a small scheme depicting the processing workflow would be helpful to the interested reader, and is common when reporting single-particle cryo-EM results.

Minor points:

- In Figure 1, the iron-sulfur cluster is labeled 3Fe-4S while in line 112 it is referred to as Fe3-S4 – this should be consistent
- Line 66: „NTP“ should be defined.
- Line 67: For clarity, I think it should be mentioned that all these factors act in different transcription systems: TFIIS (Pol II), TFS (archaea), GreA/B (bacteria).
- Line 98: Sso and Ssh should be defined and references to the papers describing their structures as well as the structure of eukaryotic RNAPII should be added (Hirata et al., 2008, Wojtas et al., 2012, and I think Cramer et al. 2001 would be more appropriate here than Dienemann et al., 2019 used in the alignment in Figure S2)
- Paragraph 142 – 158: Stylistic issue: I found this paragraph a bit confusing at first, because it jumps back and forth between the description of the structure of RIP and how it binds to RNAP. I think it would be a bit clearer if the authors first described the structure of RIP and its comparison to ORF131, and then how it binds to RNAP and that this is mediated by its unique C-tail which is not conserved and explains the function of RIP.
- Line 158: Figure 2B should be referenced here.

- Line 178: Reference 18 seems misplaced, as it does not show a mutagenesis analysis of RIP
- Reference 9 and 10 are identical
- Line 192: Reference 27 should be placed after “solution”
- Line 204: “closed” typo
- Line 298: Reference 22 should be cited here.
- Line 323: “of” should be “by”.
- Line 384: Reference 18 seems misplaced here, as it does not show that minimal PICs lacking TFE are more sensitive to RIP. Should be replaced by reference 10.
- Line 386: Typo “reliable”
- Line 388: Typo “and and”
- Lines 390 and 391: MAF1 and MAF-1 – should be consistent
- Line 464: RPR9 should be RPB9
- Lines 468-470: “A12” should be “RPA12”, and I think this sentence needs to be rearranged in order to convey that the “inside funnel” state is associated with termination.
- Line 505: “that” seems misplaced
- Figure S7: The labeling seems incorrect: The red density is TFIIS and the blue density is Rpb9
- The authors should cite the respective papers when they mention published structures and PDB codes in text or figure legends.

A final comment:

- In lines 481 – 485 the authors speculate about the evolution of RNAP subunits and factors that bind near the rim of RNAP. While beyond the scope of the discussion in this paper, it may be of interest that the poxviral RNAP, a viral multi-subunit RNAP related to cellular, bacterial and archaeal RNAPs, also contains a subunit (Rpo30) that binds to the rim. This subunit has a ZR-domain that may enter the pore in a TFIIS-like fashion to promote transcript cleavage and in addition has a phosphorylated tail which can occupy the DNA and RNA binding sites in the active site to inhibit the viral RNAP. Thus, this factor binds to a similar location as the discussed factors, and combines both stimulatory and inhibitory features in one protein.

Reviewer #2 (Remarks to the Author):

Pilotto et al. describe the high-resolution cryo-EM structures of the archaeon *Sulfolobus acidocaldarius* RNA polymerase (RNAP) and its chemically crosslinked complexes with two distinct transcriptional regulators, an ATV virus-encoded inhibitor RIP, and an infection-induced host cell negative regulator TFS4. The comparative structural analysis revealed that the modes of action of RIP and TFS4 entail a complex combination of steric occlusion and allosteric effects on RNAP. ATV RIP (a paralog of the viral P131/ORF131 capsid protein) folds into a compact single-domain α -helical bundle. It snugly fits inside the RNAP DNA-binding (primary) channel held by tight interactions with the clamp helices and the rudder. Notably, the position of the C-terminal tail of RIP mimics that of the basal initiation factor TFB (a homolog of eukaryotic TFIIB) B-linker region in the preinitiation complex (PIC). The authors conclude that RIP acts as a classic competitive inhibitor blocking the binding sites for the DNA (and RNA/DNA hybrid) and TFB. Unlike RIP, TFS4 comprises two Zn-ribbon domains (ZRN and ZRC) connected by a flexible linker similar to the structures of A12 and C11 subunits of eukaryotic RNA Pol I and Pol III, respectively. Accordingly, TFS4 anchors to

RNAP via its ZRN, placed at the lobe domain near the edge of the primary channel wall. However, the position of the TFS4 linker and ZRC domains differs from that of AC12 and C11. The TFS4 linker is wedged between the upper jaw and the lobe domains of RNAP, whereas ZRC binds to the rim helices of the NTPs-entry port funnel (secondary channel) without reaching the active site. The authors propose that such placement of TFS4 induces the opening of the DNA-binding channel and causes the disruption of the bridge helix and trigger loop elements of the active center, thus resulting in RNAP inactivation. Because similar structural perturbations have been reported for bacterial RNAPs in complex with a negative regulator Gfh1, the authors conclude that the allosteric inhibition of the RNAP is evolutionary conserved across all domains of life.

The results presented in this paper are fascinating; they offer new insights into the mechanisms of RNAP inhibition and transcription regulation and thus provide an important contribution to the field. I believe the manuscript can be published in *Nature Communications* after a moderate revision. My primary concern is the validity of the authors' conclusion that TFS4 (and bacterial Gfh1) acts primarily allosterically by locking the jaw/lobe/clamp domains in specific (open) conformation "exploiting the RNAP structural flexibility". Though plausible, this assertion is based on the observed RNAP/TFS4 conformation artificially fixed by BS3 crosslinking. The "wedging" effect of the TFS4 linker on the upper jaw/lobe domains alone is insufficient to destabilize the PIC, as the TFS4 mutant carrying K76/K77/K78 triple Ala substitution is functionally inactive (Fouqueau et al., 2017). The inhibitory action of TFS4 could result simply from the displacement of the trigger loop from the active site. Bacterial DksA, also acting through the secondary channel, destabilizes the initiation complexes without any significant perturbation of the clamp domains or DNA-binding channel opening. Additionally, the described RNAP/TFS4 structure does not reveal the interacting partner(s) in RNAP for the three TFS4-ZRC Lys residues or explain the mechanism by which they could play such a critical role in TFS4 activity. The authors should elaborate on these issues. My other critical comments (mostly minor) and suggestions are summarized below.

1. Introduction, page 2 (lines 64-66), and Discussion, page 20 (lines 421-425). Depending on promoter context, DksA (alone and together with ppGpp) inhibits and stimulates transcription. It's assumed that DksA acts allosterically by destabilizing RNAP-DNA interactions in early initiation complexes en route to RPo formation. However, the exact molecular mechanism of action is currently unknown.
2. Results, page 3, 1st para. The authors should provide the rationale for using the chemical crosslinking for cryo-EM studies of RNAP and its complexes with inhibitory factors and discuss the advantages and limitations of this approach.
3. Page 8, lines 164-174. It seems strange that the binding of RIP in the primary RNAP channel between the Rpo1' clamp, and the Rpo2 protrusion and lobe motifs, does not cause any global or even local perturbation of RNAP structure. The authors should comment on this observation.
4. Page 10 (lines 223-224). Figure 3D does not look convincing. The signal from the Cy3-labeled DNA/RNA scaffold has very low intensity, and the effect of RIP on TEC is hardly visible. It appears that RNAP makes a tripartite complex with RIP and DNA/RNA scaffold.
5. Page 13 (lines 274-280). The depiction of the TFS4 structure in Figure 4A-C and how its ZRN, linker, and ZRC domains interact with the upper jaw, lobe, and rim helices of RNAP is vague and unclear. I suggest showing the ribbon structures of TFS4 and its most relevant eukaryotic homologs, subunits A12 and C11, indicating the positions of functionally important acidic and basic (especially K76/K77/K78) residues. The domain organization and RNAP-binding properties of TFIIS are different from TFS4, A12, and C11, so its structure may be shown in the supplementary Figure as a reference. The supplementary movie (S1) is

hard to follow. It lacks information on the location of key structural elements (the catalytic center, bridge helix, trigger, jaw, and clamp domains) in the structure. I suggest showing these elements in different colors or shades of gray.

6. Page 13 (lines 288-289) and page 14 (lines 305-307). The authors state that the TFS4 ZRC domain clashes with and displaces the trigger loop without occluding the pore. The structure does not actually show that. Moreover, the trigger loop is intrinsically flexible and can assume various conformations and spatial positions in the funnel. Thus, TFS4 ZRC may simply prevent the trigger loop from closing.

7. Page 14 (lines 298-289). The authors hypothesize that a cluster of positively charged residues (R57/R65/K76/K77/K78/R82) is responsible for the initial TFS4 binding inside the negatively charged channel. However, no evidence for such a possibility was provided. The authors should provide a structural view of the channel showing the local surface charge distribution to support this idea. Also, the K76/K77/K77 residues do not seem to contact any sites in RNAP, yet their alanine substitution abolishes TFS4 activity. The authors should explain this discrepancy.

8. Page 16 (line 323). Do the authors mean: "...increasing the K_m for substrate NTPs"? Lowering the K_m for NTPs would be equivalent to increasing the binding affinity to the substrate.

9. Discussion, page 19 (lines 391-392). Gp2 does not occlude the binding site for the sigma factor to RNAP clamp and rudder (as can be seen in Fig. 6A). Instead, it blocks the interaction between the downstream DNA and the β' jaw domain, which is essential for the promoter open complex formation.

10. Page 30 (line 718-719). Ref. 9 is the same as ref. 10.

11. Page 31 (line 749-750). Ref. 22 should be updated.

12. Page 43, Supplementary Figure S7. Incorrect depiction of RPB9 and TFIIIS in panel S7b. Also, The position and orientation of ZRC domains of TFS-related proteins in the secondary channel (the funnel) shown in panel S7e is nearly impossible hard to see. I suggest the authors redraw this Figure to show the difference between the binding of various ZRC domains.

REVIEWER COMMENTS

Reviewer #1 (Remarks to the Author):

....

The results presented here provide detailed molecular mechanisms for the inhibition of archaeal RNAP by RIP and TFS4. These data are important and novel findings that are not only of interest to those studying archaea, but also for the broad field of transcription and its regulation. The conclusions are well supported by the data, and the structure analysis by single-particle cryo-EM is state-of-the-art. I have no technical concerns, and no further experiments are needed to support the claims. In conclusion, I think this manuscript should be published after addressing some open questions and comments outlined below, which mainly concern the presentation and discussion of the data.

Major points:

1. The authors describe a novel zinc binding site in Rpo1', which they term Zn3. In Figure S3a it appears that the zinc ion is coordinated by only three residues (Cys, Cys, His). A coordination number of 3 is extremely rare for zinc in protein structures (PMID: 24059258). Is there a fourth coordination partner in vicinity? If not, is there any additional evidence in the literature that this motif indeed binds zinc?

The zinc ion is coordinated by four ligands: the carbonyl group of Arg573, and the side chains of Cys575, Cys580, and His582. The carbonyl group is the unusual, weak, and unexpected ligand, which might account for the higher flexibility in the local area of the map. This zinc finger is conserved only in Crenarchaea (figure S2, now Supplementary figure 2), the coordination system is now properly described in the figure legend.

2. Lines 111 – 118: In this paragraph, the authors describe the structure of apo-RNAP (determined at 2.9 Å) and that the high quality of the cryo-EM reconstruction allowed modeling of various structural features. However, in some of the referenced figures (S3), the 2.6 Å cryo-EM density from the subsequently described RNAP-TFS4 dataset is shown to emphasize these features. Which density is shown in Figure 1B and 1C? The authors should clearly state this to avoid confusion.

For figure 1B and C (now Figure 1b and c) we used the EM map of the apo-RNAP. We corrected the corresponding figure legend to clarify this point.

3. Lines 115 – 118: This sentence seems confusing, because it has not been stated how the initial model of Rpo8 was obtained, and it is therefore not clear what was rebuilt. I would suggest to include a half-sentence along the lines of "Compared to previous archaeal RNAP structures, ..." to clarify this.

We improved this section according to the reviewer's suggestion and additional details are available in the Supplementary Methods. The revised text is now reported on lines 100 - 104.

4. Lines 181 – 187: The sentence starting with "ATV has a broad host range..." should be more

specific: I assume it is not the virus that inhibits *M. jannaschii* RNAP, but purified RIP. Figure S6d shows that several of the critical residues appear to be conserved in *S. cerevisiae* Pol II. Is it known whether RIP also inhibits eukaryotic RNAPs? Are homologues of RIP encoded by viruses that infect eukaryotes?

Lines 167 - 170: We improved the clarity by rephrasing this sentence: *'This is in good agreement with the observations that (i) RIP also inhibits the euryarchaeal M. jannaschii RNAP in vitro but not the [...]*

The sequence alignment in the revised figure S6d (now Supplementary figure 7d) includes both RPA1 and RPC1 and shows that the residues involved in the hydrogen bond network are not conserved - making RIP unlikely to inhibit RNAPI and III. Unfortunately, as we do not have *in vitro* transcription assays for RNAPI, II and III in our lab, we cannot test whether RIP inhibits eukaryotic RNAPs experimentally to unequivocally prove this point.

Neither RIP nor ORF131 has homologs in the bacterial and eukaryotic domains of life.

5. Figure 3A: The description of what is shown is not entirely clear: Are the polymerase model and RIP from the structure described in this paper and the DNA, TFIIB and TBP from δ gyk? Or was RIP from the current structure simply overlaid with δ gyk? It also appears as if TFIIA (chains U and V) were omitted, which should be stated in the legend. The same applies to panel C and F.

The RNAP and RIP in the superimposition model shown in figure 3 are based on the RNAP-RIP complex, while the TBP, TFIIB and DNA are based on the eukaryotic PIC. Additional factors that are not conserved in archaea such as TFIIA was not included in panels a and c. Likewise, for clarity, we do not show Spt4/5 in panel f. We have corrected the figure legend to clarify.

6. The authors show that RIP occupies the same binding site as the DNA and TFIIB-like factors and that adopts a "similar binding mode to the RNAP rudder" (line 210). Does this refer to conserved interactions on the amino acid level with the RNAP or similar structural motifs, or does it merely occupy the same binding site?

The text has been corrected accordingly. The sequence is not conserved, but RIP and TFB share the same binding site and structural features. The revised text corresponds now to lines 192 - 196.

7. Lines 226 – 228: The authors conclude that their data support a competitive binding mode of RIP and the DNA/RNA nucleic acid scaffold, which is also supported by the structure. However, previous data by the authors (reference 10) indicated that RIP and the nucleic acid do not compete with each other for binding to RNAP. Is there an obvious explanation for these results?

Lines 205 - 215: The competition is likely ineffective/incomplete and dependent on the exact experimental conditions *in vitro*. RIP inhibits elongation to a much lesser extent than initiation (Sheppard et al., Nature Comm 2016), which suggests that RIP interferes with but is not able

to displace the DNA from the DNA-binding channel entirely once the TEC has formed, as shown in Figure 3d.

8. Figure 3E and lines 235-262: The experiment showing that RIP is apparently exclusively expressed during late stages of infection is very striking. Is it known what other ATV genes follow a similar expression pattern? Is this the stage when capsid proteins are also expressed? Are there any indications that RIP could act in concert with other viral factors?

There is no comprehensive analysis of temporal gene expression in ATV. However, capsid proteins are usually expressed during the very late stages of infection, with a similar temporal expression profile of RIP. There is no evidence in the literature that RIP interacts with other viral or host factors.

9. Figure 4: I had a hard time correlating the description of results (paragraph lines 264 – 307) with this figure, and I think it could be more intuitively understood with a few rearrangements. First, I found it difficult to decipher which parts belong to TFS4 and which parts belong to RNAP in panel A and C.

I think the figure would benefit from a schematic depiction of TFS4 (for example as bar representing the primary sequence), with domain annotation and colored accordingly as in panels A and C. For panel A, may I suggest the authors depict structural models instead of cryo-EM density as surface? I find the latter adds relatively little information content, but makes it difficult to see details (this also holds true for Figure 2C, but there it is clearer how and where RIP binds). One possibility would be to show the RNAP model as surface rendering and slightly transparent, and only TF4S and the elements of RNAP that interact with it as cartoon or surface and coloured (for example as in another paper by this group, ref 22 Figure 7a). In combination with the above-mentioned schematic of TF4S, I believe this would make the figure much more intuitive. Regions shown enlarged in panels B and C could then possibly also be indicated by small boxes around them. Moreover, in panel C, some residues in the jaw and TFS4 are emphasized as sticks, but not labeled, and it is thus not clear where for example the important lysine cluster (K76/77/78) is located. Finally, I would suggest the authors consider swapping Figure 4D with the structural comparison in Figure S7a-d. This could then be referenced at the end of the sentence in line 288.

We have improved figure 4 and now included a colour code that makes it as clear as possible to discern between elements of RNAP and TFS4. During the 'map segmentation' step used to colour the domains, the map was smoothed which generated a lower quality map. This issue was overcome using the 'colour zone' tool in Chimera. This tool has been applied in the maps shown in figure 1a and 2b as well, which are now of better quality. We had previously prepared the figure showing the model (as ribbon representation enveloped within a semi-transparent surface) as suggested by reviewer-1, but this did not really improve the clarity of the figure, which is why we settled on showing the map and the structural model of TFS4 separately. This model also emphasises the interaction network of H-bonds, and the position of the three lysine residues. We appreciate the reviewer's comment regarding our previous work, however the interaction network is more extensive and cannot easily be displayed using the suggested style. The electrostatic potential surfaces have been updated (also considering the comments of the reviewer-2) and included as separate panels, as suggested by reviewer-

1. We do feel that the sequence alignment in figure 4e contributes to provide important information, i. e. that TFS4 is a *bona fide* paralog of TFS1, TFIIS and the paralogous eukaryotic RNAP subunits, as well as highlighting the positions of the acid loop and lysine cluster. Following the suggestion of reviewer-1, the schematic of the domain organization has been colour coded to help the reader to understand the colour code used in the structure representation. The aim of figure 4 is to show the binding mode of TFS4 to RNAP, and the relation to cleavage factors and RNAP subunits. The commonalities and differences with TFS4 homologs are described later in figure 7 and S7 (now Supplementary figure 8).

10. This group has previously shown that a lysine cluster in TF4S (K76/77/78) is critical for its function in RNAP inhibition (reference 22). Does the RNAP-TF4S structure provide direct evidence for the role of these residues? In Figure 4B and C, they appear to be depicted no interactions are obvious. In lines 298 – 302, the authors speculate that these residues may play a role during initial binding of TFS4 to RNAP, and that TFS4 undergoes a remodeling during binding to RNAP. Is this somehow inferred from the structural data presented here, or is it a hypothesis? In the latter case, I think it should rather be moved to the discussion.

Yes indeed, the TFS4-bound RNAP structure does not provide any direct information about the interactions of K76/77/78 and the stepwise binding model is inferred from the structure. Following the recommendation of the reviewer, we have rewritten this part (now lines 261 - 269) and moved the hypothesis we formulated from the results to the discussion part of the manuscript (now lines 436 - 484).

11. Organizational issue: In lines 288 – 289 the authors mention that TFS4 clashes with the trigger loop without occluding the pore, but this is only explained in the next section (from lines 321 on) and no reference is given to Figure 5 where it is depicted. Therefore, this statement stands isolated here. Similarly, the section in lines 302 – 307 discusses the similarities and differences between TFS4 and other rim-binding factors and emphasizes that TFS4 is the first example of a zinc ribbon domain that clashes with the trigger loop and induces conformational changes. However, this is only described in the section that follows. I think both the first statement (lines 288 - 289), and the concluding remarks (302 – 307) would be more appropriately placed in the following section (lines 321 and onward). This way, the first section on TF4S would focus on how the factor binds to RNAP, while the second section would describe its mechanism of inhibition.

We have reorganised this paragraph 'the TFS4 cleavage factor paralog evolved into a RNAP inhibitor' to improve the logical flow of arguments and the clarity of the text. All discussion of the displaced trigger loop and the allosteric inhibition is now moved into the next paragraph 'Allosteric modus operandi of TFS4' as recommended by reviewer-1.

12. Lines 324-326: The authors state that the binding sites of TFS4 do not overlap with those of initiation or elongation factors. However, from my understanding and the depictions in Figure S7a-d, the binding site of TFS4 does overlap with that of RPA12 / TFIIS / RPC10 in other transcription systems and likely with that of TFS1 in the archaeal system (reference 22). Could the authors clarify this?

Sorry, we had general transcription factors in mind that are essential for transcription initiation. Lines 324-326, now 274 - 276, have been rephrased, as TFS4 and TFS1 binding sites are extremely likely to overlap, although the actual structure and binding mode for TFS1 has not been solved yet.

13. Since the authors describe in the methods that specific experimental approaches were employed to overcome particle orientation bias (graphene oxide, tilting of the stage), I would strongly encourage them to include angular distribution plots in Figures S1, S4 and S5 to demonstrate the outcome of these attempts. In addition, a small scheme depicting the processing workflow would be helpful to the interested reader, and is common when reporting single-particle cryo-EM results.

The angular distribution plots have been added to each figure.

However, we did not prepare a workflow for the processing since we did not detect any additional species even at very low resolution (as explained in Methods). We simply do not have additional maps or information to show other than 2D projections and the EM map, which are already included in the supplementary figures. The processing removed 'junk' and bad particles, and the tilt means just that some particles were at very high or low defocus, lowering the quality of the dataset, thus, removed during the processing. The pruning of the TFS4 dataset did not provide any significant improvement. The Phenix autosharpen program, which removes anisotropy when present, worked very well on our un-pruned map, so that we did not notice any relevant difference between the pruned and un-pruned map. However, we decided to keep the pruned map since it did not affect the resolution.

Minor points:

- In Figure 1, the iron-sulfur cluster is labeled 3Fe-4S while in line 112 it is referred to as Fe3-S4 – this should be consistent

Corrected (now line 96)

- Line 66: „NTP“ should be defined.

Corrected (now line 63)

- Line 67: For clarity, I think it should be mentioned that all these factors act in different transcription systems: TFIIIS (Pol II), TFS (archaea), GreA/B (bacteria).

Updated (now lines 64 - 65)

- Line 98: Sso and Ssh should be defined and references to the papers describing their structures as well as the structure of eukaryotic RNAPII should be added (Hirata et al., 2008, Wojtas et al., 2012, and I think Cramer et al. 2001 would be more appropriate here than Dienemann et al., 2019 used in the alignment in Figure S2)

References added in the main text, now lines 93 - 94.

We indeed used the Cramer et al 2001 paper as reference for the domain assignment, however the superposition was carried out with the structure from Dienemann and co-workers since the Cramer's 2001 structure lacks of the RPB4/7 subunits (Rpo4/7 in archaea).

- Paragraph 142 – 158: Stylistic issue: I found this paragraph a bit confusing at first, because it jumps back and forth between the description of the structure of RIP and how it binds to RNAP. I think it would be a bit clearer if the authors first described the structure of RIP and its comparison to ORF131, and then how it binds to RNAP and that this is mediated by its unique C-tail which is not conserved and explains the function of RIP.

We moved the last sentence earlier in the paragraph (now lines 124 - 127) to ensure the logic of the description as suggested by the reviewer.

- Line 158: Figure 2B should be referenced here.

Added as Figure 2a (now line 127)

- Line 178: Reference 18 seems misplaced, as it does not show a mutagenesis analysis of RIP

Both corrected. Reference 18 removed from line 163; correct reference 7, Sheppard 2016, inserted in line 170.

- Reference 9 and 10 are identical

Corrected

- Line 192: Reference 27 should be placed after "solution"

Now line 180: Reference 27 was misplaced. The data reported referred to Sheppard 2016 (reference 7). Corrected.

- Line 204: "closed" typo

Now line 189: corrected

- Line 298: Reference 22 should be cited here.

This part of the paragraph '*The TFS4 is a cleavage factor paralog evolved into a RNAP inhibitor*' has been moved and rewritten in the Discussion section, the sentence corresponding to line 298 is now line 476 and it has been correctly referenced with Fouqueau 2017 (reference 16) as suggested by the reviewer.

- Line 323: "of" should be "by".

Now line 273 has been corrected

- Line 384: Reference 18 seems misplaced here, as it does not show that minimal PICs lacking TFE are more sensitive to RIP. Should be replaced by reference 10.

Now line 413 has been corrected with reference 7 (Sheppard 2016)

- Line 386: Typo “relaiable”

Now line 415 has been corrected

- Line 388: Typo “and and”

Now line 418 has been corrected

- Lines 390 and 391: MAF1 and MAF-1 – should be consistent

Now line 421 has been corrected

- Line 464: RPR9 should be RPB9

Now line 530 has been corrected

- Lines 468-470: “A12” should be “RPA12”, and I think this sentence needs to be rearranged in order to convey that the “inside funnel” state is associated with termination.

We do agree with reviewer about RPA12 functional role. The sentence has been rephrased and moved to lines 483 - 484.

- Line 505: “that” seems misplaced

Now line 578 has been corrected

- Figure S7: The labeling seems incorrect: The red density is TFIS and the blue density is Rpb9

Corrected (now Supplementary figure 8)

- The authors should cite the respective papers when they mention published structures and PDB codes in text or figure legends.

All updated throughout the text and the figure legends.

A final comment:

- In lines 481 – 485 the authors speculate about the evolution of RNAP subunits and factors that bind near the rim of RNAP. While beyond the scope of the discussion in this paper, it may be of interest that the poxviral RNAP, a viral multi-subunit RNAP related to cellular, bacterial and archaeal RNAPs, also contains a subunit (Rpo30) that binds to the rim. This subunit has a

ZR-domain that may enter the pore in a TFIS-like fashion to promote transcript cleavage and in addition has a phosphorylated tail which can occupy the DNA and RNA binding sites in the active site to inhibit the viral RNAP. Thus, this factor binds to a similar location as the discussed factors, and combines both stimulatory and inhibitory features in one protein.

We appreciated reviewer observation and because we are interested in the structure and function of a similar transcription machinery (ASFV) we compared Rpo30 and the viral RNAP with archaeal and eukaryotic RNAPs. The structure of VACV RNAP is quite different from other DPBB RNAPs. Most of the differences involved the solvent exposed surface, which in turn affect the binding mode and structure of various transcription factors, included Rpo30. More specifically, the Rpo30 N-terminal domain resembles the alpha helical bundle of TFIS although sequence and size of the domain vary greatly. However, this domain is not present in TFS4, hence less relevant for comparison or discussion. The Rpo30 C-terminal domain binds deep inside the funnel, but, as also mentioned by the reviewer, it seems structurally dissimilar from the classical zinc ribbon domain making difficult a direct comparison with TFS4 with which it does not share any structural feature and binding site/mode.

Reviewer #2 (Remarks to the Author):

...

The results presented in this paper are fascinating; they offer new insights into the mechanisms of RNAP inhibition and transcription regulation and thus provide an important contribution to the field. I believe the manuscript can be published in Nature Communications after a moderate revision.

My primary concern is the validity of the authors' conclusion that TFS4 (and bacterial Gfh1) acts primarily allosterically by locking the jaw/lobe/clamp domains in specific (open) conformation "exploiting the RNAP structural flexibility". Though plausible, this assertion is based on the observed RNAP/TFS4 conformation artificially fixed by BS3 crosslinking.

Crosslinking is a standard procedure to stabilize higher-order complexes and avoid dissociation due to the low concentrations used in cryo-EM. There is no evidence that this procedure unduly produces artificial conformations other than oligomerization and aggregation phenomena. BS3 is added after the complex is formed, and not before, where it could interfere with TFS4 binding. Importantly, we did not detect other particle species in our dataset, as explained in the Methods. Such high reproducibility, together with the high resolution achieved, makes it unlikely that our structural information is biased by artificial phenomena due to crosslinking.

The "wedging" effect of the TFS4 linker on the upper jaw/lobe domains alone is insufficient to destabilize the PIC, as the TFS4 mutant carrying K76/K77/K78 triple Ala substitution is functionally inactive (Fouqueau et al., 2017). The inhibitory action of TFS4 could result simply from the displacement of the trigger loop from the active site.

Multiple interactions enable the binding of TFS4 on the RNAP and interfering with the interactions facilitated by the ZR^C lysine residues may impair the recruitment and binding of

the whole factor to an extent where the 'wedging' cannot occur. We agree that the displacement of the trigger loop could explain the inhibition of catalysis, but this is not sufficient to explain PIC destabilization. In contrast, changes in the width of the DNA binding channel and its conformationally locked state is likely to destabilize the PIC.

Bacterial DksA, also acting through the secondary channel, destabilizes the initiation complexes without any significant perturbation of the clamp domains or DNA-binding channel opening.

We agree, the similarity between DksA and TFS4 action is very intriguing, as we discuss in the article, but that does not imply that their binding modes and effects on complex integrity, stability and catalysis are identical. In that respect, despite a similar binding site, DksA employs a mechanism of inhibition that differs from TFS4.

Additionally, the described RNAP/TFS4 structure does not reveal the interacting partner(s) in RNAP for the three TFS4-ZRC Lys residues or explain the mechanism by which they could play such a critical role in TFS4 activity. The authors should elaborate on these issues.

This is an interesting result that our structure cannot fully explain. The K76/K77/K77 residues are solvent exposed in our structure and make no direct contacts to RNAP. In our manuscript, we propose a step wise binding model that includes an early binding event that is facilitated by ZRC lysine residues making electrostatic contacts with a negatively charged patch in the RNAP funnel. However, as we cannot provide any evidence for the specific interactions in the early binding state, this model is inferred from structure and biochemical experiments, and thus included later in the discussion section of the paper where we elaborate on the matter (lines 436 - 484).

My other critical comments (mostly minor) and suggestions are summarized below.

1. Introduction, page 2 (lines 64-66), and Discussion, page 20 (lines 421-425). Depending on promoter context, DksA (alone and together with ppGpp) inhibits and stimulates transcription. It's assumed that DksA acts allosterically by destabilizing RNAP-DNA interactions in early initiation complexes en route to RPo formation. However, the exact molecular mechanism of action is currently unknown.

Agreed. We cited DksA as an example but we focused our discussion on the comparison with Gfh1. We have introduced reviewer-2's suggestion by clarifying the mechanism in lines 5494 - 496.

2. Results, page 3, 1st para. The authors should provide the rationale for using the chemical crosslinking for cryo-EM studies of RNAP and its complexes with inhibitory factors and discuss the advantages and limitations of this approach.

We introduced a sentence in lines 110 - 112 to provide the rationale for using BS3, which is a mild crosslinker causing very moderate aggregation/precipitation phenomena when

incubated at 65°C, the experimental conditions used to reconstitute our hyper-thermophilic complex.

3. Page 8, lines 164-174. It seems strange that the binding of RIP in the primary RNAP channel between the Rpo1' clamp, and the Rpo2 protrusion and lobe motifs, does not cause any global or even local perturbation of RNAP structure. The authors should comment on this observation.

Yes, but a comparison of apo- and RIP-bound RNAP structures showed that RIP fits perfectly inside the channel. The extensive binding interface stabilizes one specific RNAP clamp conformation, the closed conformation, and this is the 'perturbation' of RIP at global level. This topic has been properly addressed in the Discussion on lines 400 - 407.

4. Page 10 (lines 223-224). Figure 3D does not look convincing. The signal from the Cy3-labeled DNA/RNA scaffold has very low intensity, and the effect of RIP on TEC is hardly visible. It appears that RNAP makes a tripartite complex with RIP and DNA/RNA scaffold.

The Cy3-labelled DNA signal is weaker than the ³²P signal of RIP, but clearly detectable. Furthermore, the addition of RIP clearly reduces the Cy3 TEC signal in a dose-response fashion. EMSAs of DNA-RNAP complexes could appear as doublets or even multiple bands (e. g. Blombach et al., 2015, figure 6, <https://elifesciences.org/articles/08378>), but importantly there is no evidence of new RIP-dependent bands in the EMSA in figure 3d.

5. Page 13 (lines 274-280). The depiction of the TFS4 structure in Figure 4A-C and how its ZRN, linker, and ZRC domains interact with the upper jaw, lobe, and rim helices of RNAP is vague and unclear. I suggest showing the ribbon structures of TFS4 and its most relevant eukaryotic homologs, subunits A12 and C11, indicating the positions of functionally important acidic and basic (especially K76/K77/K78) residues. The domain organization and RNAP-binding properties of TFIIIS are different from TFS4, A12, and C11, so its structure may be shown in the supplementary Figure as a reference. The supplementary movie (S1) is hard to follow. It lacks information on the location of key structural elements (the catalytic center, bridge helix, trigger, jaw, and clamp domains) in the structure. I suggest showing these elements in different colors or shades of gray.

We have redesigned Figure 4 according to reviewer-1 and -2 suggestions, see page 4 point 10.

The movie has been updated according to reviewer-2's recommendation to enhance the clarity.

6. Page 13 (lines 288-289) and page 14 (lines 305-307). The authors state that the TFS4 ZRC domain clashes with and displaces the trigger loop without occluding the pore. The structure does not actually show that. Moreover, the trigger loop is intrinsically flexible and can assume various conformations and spatial positions in the funnel. Thus, TFS4 ZRC may simply prevent the trigger loop from closing.

We politely beg to differ with reviewer-2. Our structure shows in figures 5c that the TFS4 ZRC^C domain clashes with the trigger loop in its 'closed' conformation, the conformation it adopts in our apo-RNAP structure. Obviously, the trigger loop is still present, so we agree with the reviewer that the ZRC domain will prevent trigger loop 'closure', however, the clash is responsible for the loss of density in our RNAP-TFS4 map. This has been addressed in lines 378 - 382.

The RNAP trigger loop is not structurally resolved in several other published eukaryotic RNAP structures, as reviewer-2 states this is likely due to its inherent flexibility. But none of the eukaryotic paralogs of the TFS4 ZRC domains is sufficiently close to the trigger loop to explain the loss of density in those maps as shown in figure 7b and supplementary figure 8e.

7. Page 14 (lines 298-289). The authors hypothesize that a cluster of positively charged residues (R57/R65/K76/K77/K78/R82) is responsible for the initial TFS4 binding inside the negatively charged channel. However, no evidence for such a possibility was provided. The authors should provide a structural view of the channel showing the local surface charge distribution to support this idea. Also, the K76/K77/K77 residues do not seem to contact any sites in RNAP, yet their alanine substitution abolishes TFS4 activity. The authors should explain this discrepancy.

We have improved figure 4 to include the electrostatic potential surface of the funnel (panel d) as suggested by the reviewer.

The K76/K77/K77 residues are indeed solvent exposed in our structure and make no direct contacts to RNAP. In our manuscript, we propose a step wise binding model that includes an early binding event that is facilitated by ZRC lysine residues making electrostatic contacts with a negatively charged patch in the RNAP funnel. However, as we cannot provide any evidence for the specific interactions in the early binding state, this model is inferred from structure and biochemical experiments, and thus included later in the discussion section of the paper where we elaborate on the matter (lines 436 - 484).

8. Page 16 (line 323). Do the authors mean: "...increasing the Km for substrate NTPs"? Lowering the Km for NTPs would be equivalent to increasing the binding affinity to the substrate.

Mistake corrected, now in line 273.

9. Discussion, page 19 (lines 391-392). Gp2 does not occlude the binding site for the sigma factor to RNAP clamp and rudder (as can be seen in Fig. 6A). Instead, it blocks the interaction between the downstream DNA and the β' jaw domain, which is essential for the promoter open complex formation.

We do agree with the reviewer's comment and we improved the clarity of the text now on lines 416 - 419.

10. Page 30 (line 718-719). Ref. 9 is the same as ref. 10.

Corrected

11. Page 31 (line 749-750). Ref. 22 should be updated.

Updated

12. Page 43, Supplementary Figure S7. Incorrect depiction of RPB9 and TFIIIS in panel S7b. Also, The position and orientation of ZRC domains of TFS-related proteins in the secondary channel (the funnel) shown in panel S7e is nearly impossible hard to see. I suggest the authors redraw this Figure to show the difference between the binding of various ZRC domains.

We corrected figure S7b and modified panel (e) to highlight the differences (now Supplementary figure 8).